# KDM5D Histone Demethylase Identifies Platinum-Tolerant Head and Neck Cancer Cells Vulnerable to Mitotic Catastrophe

**DOI:** 10.3390/ijms24065310

**Published:** 2023-03-10

**Authors:** Tsung-Ming Chen, Chih-Ming Huang, Syahru Agung Setiawan, Ming-Shou Hsieh, Chih-Chi Sheen, Chi-Tai Yeh

**Affiliations:** 1Department of Otolaryngology, School of Medicine, College of Medicine, Taipei Medical University, Taipei City 11031, Taiwan; 09326@s.tmu.edu.tw; 2Department of Otolaryngology-Head and Neck Surgery, Shuang Ho Hospital, Taipei Medical University, New Taipei City 23561, Taiwan; 3Department of Otolaryngology, Taitung Mackay Memorial Hospital, Taitung City 950408, Taiwan; mmh4621@gmail.com; 4Department of Nursing, Tajen University, Pingtung 90741, Taiwan; 5International Ph.D. Program in Medicine, College of Medicine, Taipei Medical University, Taipei City 11031, Taiwan; setiawan.syahru@gmail.com; 6Department of Medical Research & Education, Shuang Ho Hospital, Taipei Medical University, New Taipei City 23561, Taiwan; 7School of Dentistry, College of Oral Medicine, Taipei Medical University, Taipei City 110, Taiwan; dr7d9eddd@gmail.com; 8Department of Dentistry, Shuang Ho Hospital, Taipei Medical University, New Taipei City, 235, Taiwan; 9Department of Periodontics, Shuang Ho Hospital, Taipei Medical University, New Taipei City 23561, Taiwan; 10Continuing Education Program of Food Biotechnology Applications, College of Science and Engineering, National Taitung University, Taitung 95092, Taiwan

**Keywords:** *KDM5D*, *AURKB*, drug-tolerant persister, cisplatin, head and neck cancer

## Abstract

Head and neck squamous cell carcinoma (HNSCC) is a major contributor to cancer incidence globally and is currently managed by surgical resection followed by adjuvant chemoradiotherapy. However, local recurrence is the major cause of mortality, indicating the emergence of drug-tolerant persister cells. A specific histone demethylase, namely lysine-specific demethylase 5D (*KDM5D*), is overexpressed in diverse types of cancers and involved in cancer cell cycle regulation. However, the role of *KDM5D* in the development of cisplatin-tolerant persister cells remains unexplored. Here, we demonstrated that *KDM5D* contributes to the development of persister cells. Aurora Kinase B (AURKB) disruption affected the vulnerability of persister cells in a mitotic catastrophe–dependent manner. Comprehensive in silico, in vitro, and in vivo experiments were performed. *KDM5D* expression was upregulated in HNSCC tumor cells, cancer stem cells, and cisplatin-resistant cells with biologically distinct signaling alterations. In an HNSCC cohort, high *KDM5D* expression was associated with a poor response to platinum treatment and early disease recurrence. *KDM5D* knockdown reduced the tolerance of persister cells to platinum agents and caused marked cell cycle deregulation, including the loss of DNA damage prevention, and abnormal mitosis-enhanced cell cycle arrest. By modulating mRNA levels of *AURKB*, *KDM5D* promoted the generation of platinum-tolerant persister cells in vitro, leading to the identification of the *KDM5D*/*AURKB* axis, which regulates cancer stemness and drug tolerance of HNSCC. Treatment with an AURKB inhibitor, namely barasertib, resulted in a lethal consequence of mitotic catastrophe in HNSCC persister cells. The cotreatment of cisplatin and barasertib suppressed tumor growth in the tumor mouse model. Thus, *KDM5D* might be involved in the development of persister cells, and *AURKB* disruption can overcome tolerance to platinum treatment in HNSCC.

## 1. Introduction

Head and neck cancer is among the most commonly diagnosed cancers globally, with head and neck squamous cell carcinoma (HNSCC) being the predominant type. In 2018, the estimated global incidence was more than 500,000 cases per year, and the mortality rate for HNSCC was approximately 50% [1,2]. Advanced disease stage, relapse, and the lack of effective treatment are the key factors contributing to mortality in HNSCC [3]. Typically, surgical resection followed by systemic platinum-containing chemotherapy and radiotherapy is used to treat HNSCC. Common platinum agents, such as cisplatin, target highly proliferative cancer cells and are frequently used in combination with other agents, including anti-epidermal growth factor receptor (EGFR) agents, cetuximab, and anti-programmed death protein 1 (PD-1) drugs; the response rates of pembrolizumab as indicated by objective indicators were found to be 36% and 43%, respectively [4,5]. Therefore, regardless of the type of combination strategies employed, platinum agents are still widely used to treat HNSCC. Furthermore, relapse in HNSCC significantly contributes to mortality [6]. Approximately 42% of patients with HNSCC who received first-line platinum-containing systemic treatment experienced recurrence [7]. These findings suggest that a residual subpopulation of cancer cells remains resistant to platinum treatment.

Cancer cells exhibit resistance to therapeutic drugs through various mechanisms. In drug-susceptible tumors, a small proportion of cells can transform into drug-tolerant persister cells (DTPCs) and contribute to the development of a drug-resistant population [8,9]. DTPCs have distinct stemness characteristics, slow-cycling profile, quiescent behavior, and diapause-state-like properties [10,11]. The diapause state represents a dormant, non-proliferating subset of cancer cells, mimicking embryonic development to prevent an exogenous insult, in this case, chemotherapy-induced cell death [12]. These distinct features can result from the selection of existing intrinsically refractory clones or adaptive transcriptional plasticity–driven mechanisms during treatment [10]. Many factors are involved in the development of DTPCs, including epigenetic modification, transcriptional regulation, metabolic remodeling, and factors related to the tumor microenvironment [13]. Epigenetic regulation and gene transcription are affected by DNA methylation and histone modification, which occur at numerous sites and are critical for cellular plasticity and treatment sensitivity during development [14,15]. In addition, drugs targeting epigenetic regulation can alter tumor sensitivity to other anticancer drugs and overcome therapeutic tolerance [16]. Therefore, the identification of DTPC subpopulations that persist after platinum treatment by examining their epigenetic perturbation can be the first key step to overcoming platinum tolerance in HNSCC.

Histone methylation can alter various biological features of tumors and serve as a potential target for eliminating treatment resistance [17,18]. Thus, several histone demethylases have gained attention because they mainly play a key role in determining sensitivity following some types of treatment and because such misregulation can be targeted to overcome the development of treatment tolerance in cancer [16,19,20]. Histone demethylases regulate biological processes, such as cell cycle control, DNA damage responses, heterochromatin formation, and pluripotency [21]. Among histone demethylases, the lysine-specific demethylase 5 (KDM5) family have attracted significant attention in cancer biology due to the acquisition of DTPCs. KDM5 family members are histone lysine demethylases that remove tri- and di-methyl marks from lysine residue (K4) of histone H3 protein (H3K4). Transcriptional regulation of the KDM5 family is either activated or repressed according to the site of methylation [22]. It has been shown, for instance, that a subset of melanoma cells with aberrant *KDM5B* survive platinum treatment by transforming into a slow-cycling persister state [23]. By controlling the chromatin marks of H3K4me3, the KDM5 family could regulate the expression of mitotic-regulating genes, Aurora Kinase B (*AURKB*) [23,24]. As a catalytic subunit of the chromosome passenger complex (CPC) during mitosis, *AURKB* facilitates chromosome alignment in metaphase and during cytokinesis [25]. Cancer cells could gain an advantage by modifying *AURKB* in a manner similar to how it functions during mitosis. Therefore, drugs targeting *AURKB* have become increasingly significant in recent years due to their potential to disrupt mitotic control of cancer cells while triggering a lethal cell death due to mitotic failure namely mitotic catastrophe [26]. Moreover, *AURKB* expression has been identified as a prognostic marker in several cancers, including oral cancer [27]. In this way, studying the role of KDM5 family members in maintaining DTPCs through exploiting certain cell cycle and mitosis controls could yield an alternative method for identifying yet eliminating cancer cells associated with treatment refractoriness.

Among all KDM5 family members, lysine-specific demethylase 5D *(KDM5D*) has received relatively less attention. *KDM5D* is frequently mutated in clear-cell renal cell carcinoma and is a major contributor to carcinogenesis [28]. *KDM5D* expression in gastric cancer cells substantially reduced these cells’ viability, implying that it may inhibit direct growth [29]. *KDM5D* deficiency results in an increase in H3K4me3 methylation, leading to DNA replication stress and genomic instability [30]. This alteration increases the level of the G2/M checkpoint regulator and modulates the activation of the Ataxia-telangiectasia- and Rad3-related protein kinase (ATR)-dependent mechanism through DNA replication stress. Thus, *KDM5D* is closely related to the epigenetic regulation of cell cycle control in cancer. However, the significance of *KDM5D* to the development of DTPCs in HNSCC remains poorly understood. Herein, we examined the putative role of *KDM5D* in orchestrating *AURKB* expression might contribute to the acquisition of DTPCs in HNSCC following platinum treatment. Platinum-tolerant persister cells of HNSCC then could be exploited by targeting *KDM5D*-associated control of cell cycle, DNA damage repair mechanism, and *AURKB*-mediated mitotic control by treating with *AURKB* inhibition, which provoked mitotic delay and ultimately resulted in mitotic catastrophe. Moreover, the clinical relevance of *KDM5D* as a potential marker of DTPCs and platinum tolerance in HNSCC patients would be determined.

## 2. Results

### 2.1. KDM5D Underlies the Relationships between Treatment Tolerance, Diapause State, and Cancer Stemness

To investigate the expression of genes contributing to associations among cisplatin chemoresistance, DTPC development, and cancer stemness in HNSCC, we performed in silico experiments by using representative microarray datasets. We used the datasets GSE9844, GSE72384, and GSE102787 that previously detailed the transcriptomic profiles of oral squamous cell carcinoma tumors, cancer stem cells (CSCs), and cisplatin-resistant cells in HNSCC. The Differentially Expressed Genes (DEGs) are highlighted in the volcano plot of each dataset. *KDM5D* was upregulated in HNSCC tumors, CSCs, and cisplatin-resistant cells in HNSCC (Figure 1A). According to the Venn diagram, approximately 13 genes were intersected among the three phenotypes, including *KDM5D* (Figure 1B). *KDM5D* was among the top 20 genes that were differentially expressed in the CSC subset of HNSCC, and the heatmap revealed the clustering of the CSC and non-CSC subsets of HNSCC (Figure 1C). Compared with all recognizable lysine demethylases belonging to the JARID or KDM family in humans, *KDM5D* was relatively highly expressed in HNSCC tumors and cisplatin-resistant cells; this finding serves as preliminary evidence that *KDM5D* can be used to identify tumorigenesis and indicate the treatment resistance of HNSCC cells (Figure 1D).

We determined the putative association of *KDM5D* expression with DTPC development and cancer stemness. The gene set enrichment analysis (GSEA) findings revealed that several crucial signaling pathways were activated or deactivated and linked to cancer stemness in HNSCC (Figure 1E). Several pathways were actively enriched, including pathways involved in oral leukoplakia formation, embryonic germ cell pathways, the WNT/β-catenin signaling pathway, and pathways involved in the deregulation of MYC/E2F1 target genes. The diapause state closely resembled the embryonic development stage. The Wnt/β-catenin pathway mediated the progression of CSC cells, and the downregulation of Myc/E2F1 signaling in CSCs cells resulted in the deceleration of the cell cycle. The deactivated pathways in CSCs cells were those involved in the suppression of the cell cycle and deactivation of mammalian target of rapamycin complex 1 (MTORC1) and unfolded protein response signaling. The findings indicated that CSC cells have quiescent features (e.g., diapause state) similar to those of embryonic cells. Furthermore, other biological mechanisms, including DNA repair, apoptosis signaling, and the protein P53 pathway, were deactivated in CSCs cells. The downregulation of transforming growth factor-beta (TGF-β) signaling in CSCs cells indicated the blockage of cellular differentiation and thus tumor initiation. Moreover, CSCs cells exhibited a lack of inflammatory and immune responses, as indicated by the deactivation of *IL6*/*JAK*/*STAT3* and *IL2*/*STAT5* signaling. Therefore, the findings suggested the activation of a diapause-like state in CSCs cells in HNSCC. The transcriptomic landscape of cells in the diapause state and DTPCs was previously described for patients with colorectal cancer. Therefore, by employing the profiling results of the TCGA-HNSC dataset, we examined the correlation between diapause enrichment and KDM5D expression. A previous study already established 14 genes that were upregulated (e.g., *PDCD4*, *ACSS1*, *HEXB*, *CTSL*, *CCNG2*, *SPRY1*, *APOE*, *ALDH6A1*) and 110 genes that were downregulated (e.g., *LDHA*, *PPA1*, *S100A6*, *ID3*, *MGMT*, *PRDX2*, *HSPE1*, *EIF2B2*, *CENPM*, *DRG2*, *PDCD5*, *EIF3B*, *CCDC28B*, *BRMS1*) during embryonal diapause state [11]. The scatter plot revealed that *KDM5D* expression was positively correlated with genes upregulated in the diapause-like state (r = 0.22, *p* = 0.0032) and negatively correlated with genes downregulated in the diapause-like state (r = −0.26, *p* = 0.0077; Figure 1F). A higher *KDM5D* expression level was associated with poorer overall survival in patients in the TCGA-HNSC dataset (*p* = 0.0049; Figure 1G). These data preliminarily suggest that *KDM5D* upregulation is involved in the associations between treatment tolerance, diapause state, and cancer stemness in HNSCC.

### 2.2. KDM5D Is Associated with Poor Clinical Outcomes in Patients with HNSCC

To determine whether KDM5D protein expression significantly affected the clinical outcomes of patients with HNSCC, we included an adequate number of patients with HNSCC (*n* = 100) and examined their respective clinical tissue specimens through immunostaining. Figure 2A presents the difference in KDM5D expression determined through immunostaining between HNSCC tissues and the adjacent normal tissues. Higher KDM5D expression was noted in HNSCC tissues than in adjacent normal epithelial tissues. The highest KDM5D expression was observed in poorly differentiated squamous cell carcinoma tissues (Figure 2B). These findings emphasize the crucial role of KDM5D expression in the tumorigenesis and aggressiveness of HNSCC. To determine whether KDM5D affected the outcome of platinum-based chemotherapy in HNSCC, we examined the response following the completion of cisplatin treatment. Treatment response was objectively assessed using the Response Evaluation Criteria in Solid Tumors criteria [31]. According to these criteria, the treatment response was divided into four categories: complete response (CR), partial response (PR), stable disease (SD), and progressive disease (PD). Patients exhibiting CR or PR were classified as responders, whereas those with SD or PD were classified as nonresponders. The patients were followed up for at least 1 year after treatment completion to determine disease recurrence or tumor relapse, and they were accordingly classified on the basis of the onset of tumor recurrence. Early recurrence was defined as any tumor recurrence occurring during the first 6 months after treatment initiation. Late recurrence was defined as any recurrence noted more than 6 months after treatment initiation. No recurrence was defined by the absence of any observation of recurrence for more than 6 months after treatment initiation [7,32]. According to the prespecified evaluation criteria, each representative HNSCC specimen with its respective KDM5D immunostaining expression was evaluated (Figure 2C). Among the patients with HNSCC, KDM5D expression was higher in the nonresponders than in the responders (Figure 2D). Moreover, among the 70 patients who responded to platinum-based chemotherapy, higher KDM5D expression was observed in those with early disease recurrence than in those with late or no tumor recurrence (Figure 2E). Results on the association between KDM5D and clinical treatment outcomes are presented in Table 1. Higher KDM5D expression in the patients with HNSCC was significantly associated with both poorer chemotherapy response (*p* = 0.004) and early tumor recurrence (*p* = 0.012). These findings indicate that KDM5D expression affects the treatment response and tumor recurrence following chemotherapy. Moreover, the clinicopathological data suggest that KDM5D contributes to the emergence of specific tumor cell subsets after chemotherapy, namely platinum-tolerant persister HNSCC cells.

### 2.3. Diapause State in Persister Cells Is Enriched by KDM5D/AURKB Coexpression

Tumors can significantly vary in terms of genetic and nongenetic factors between geographical regions or between different progression stages; this phenomenon is known as intratumor heterogeneity. Intratumor heterogeneity may affect key signaling pathways that regulate the growth of cancer cells, drive phenotypic diversity, and cause resistance to cancer treatment. Single-cell sequencing can be performed to determine the gene expression profiles of single cells and explore intratumor heterogeneity under specific circumstances. To determine the intratumor heterogeneity of HNSCC cells in terms of their tolerance persister phenotype, we examined a previously described representative single-cell profile of HNSCC. A single-cell transcriptomic profile dataset reported by Puram et al., with the code GSE103322, was selected; this dataset comprised data on eight distinct clusters of tumor cells (Figure 3A). Several well-known tumor markers for HNSCC were highly expressed in almost every cell cluster, including Keratin 6A (*KRT6A*), Keratin 14 (*KRT14*), and Cadherin-1 (*CDH1*) (Figure 3B). An established cancer stemness marker was used in the present study, namely *Aldehyde Dehydrogenase 1 Family Member A3* (*ALDH1A3*) since this Aldehyde Dehydrogenase isoenzyme has also recently been identified to be enriched in drug-tolerant persister cancer cells, cisplatin resistance, and radioresistance of head and neck cancer [33,34,35]. Here, we noted a weak expression of specific cancer stem cell markers and DTPC markers, such as *ALDH1A3*, in some clusters, such as clusters 1, 3, and 5 (Figure 3C). Relatively overall weak expression of cancer stem cell markers reflected the sampling approach of the dataset, which was derived from the fresh biopsy pretreatment HNSCC tumors without further isolation or enrichment of the CSC subset. Furthermore, the expression of *ALDH1A3*, *KDM5D* and *AURKB* was noted in each cluster (Figure 3C). We selected *AURKB* because it regulates cell cycle checkpoint and DNA damage responses, particularly by inhibiting p53 and causing chemoresistance in cancer cells [36]. This finding is consistent with our previous result that CSCs in HNSCC deregulated the DNA damage repair pathway and p53 signaling pathway (Figure 1E). The expression of KDM5D was significantly correlated with that of AURKB (r = 0.75, *p* = 0.012). The expression of *AURKB* was significantly correlated with that of *ALDH1A3* (r = 0.38, *p* = 0.035; Figure 3D). These findings might indicate the presence of *KDM5D* and *AURKB* interaction to promote *ALDH1A3*-mediated cancer stemness in HNSCC.

To confirm our speculation regarding the association between KDM5D expression and the diapause state, several markers that were upregulated during diapause were selected from previous reports, such as Cyclin D1 (*CCND1*), Fas Cell Surface Death Receptor (*FAS*), and Aldehyde Dehydrogenase 6 Family Member A1 (*ALDH6A1*) [11]. The level of expression of those markers was also shown in each cluster (Figure 3B,C). While overlap expression between *KDM5D* and these markers was not entirely evident, correlation analysis resulted in significant associations between *KDM5D* and *CCND1*, *FAS*, and *ALDH6A1* (Figure 3E). Moreover, the relatively abundant expression of KDM5D and ALDH1A3 in cell clusters 1, 3, and 5 overlapped with a portion of clusters upregulating diapause gene signatures (Diapause_UP). Moreover, these clusters of cells (no. 1, 3, and 5) co-existed with clusters that minimally expressed genes that were consistently deactivated during diapause (Diapause_DOWN) (Figure 3F). These findings indicated that the clusters contained a substantial number of cells in the diapause state, a key feature of DTPCs in HNSCC. Besides, those clusters of cells exhibited features similar to those of cells in the diapause state, including the activation of nuclear factor erythroid 2-related factor 2 (NRF2), glutathione, drug metabolism, glycolysis, and epithelial-mesenchymal transition pathways (Figure 3G). Overall, the findings provide preliminary evidence that *KDM5D*/*AURKB* axis activation is associated with the predomination of the persister cells cluster, which was also enriched, by *ALDH1A3* expression and the activation of the diapause state.

### 2.4. KDM5D Promoted Persister Cell Development by Modulating AURKB Expression

To determine the properties of platinum-tolerant persister cells in HNSCC, we generated an in vitro HNSCC cell line model. As presented in Figure 4A, after short-term cisplatin treatment followed by a period of no drug exposure, the HNSCC cells were relatively viable and exhibited increased tolerance (Figure 4A). The platinum-tolerant cell lines were then used as a basic model for dissecting the basic molecular mechanism of drug tolerance acquisition. Here, *ALDH1A3* was again being used to remark enrichment of drug-tolerant persister cancer cells, cisplatin resistance, and quiescent population resembling diapause state [33,34,35,37]. The quantitative polymerase chain reaction revealed relatively high expression levels of *KDM5D*, *ALDH1A3*, and *AURKB* in wild-type platinum-tolerant SAS (PT-SAS) and FaDu (PT-FaDU) cells (Figure 4B). According to previous in silico findings through multiple sets of transcriptomic profiling and our speculation regarding the putative role of the *KDM5D*/*AURKB* axis in contributing to drug-tolerant persister cells, *KDM5D* silencing was performed in platinum-tolerant HNSCC cells and determine the downstream modulation by *KDM5D*. As expected, the knockdown of *KDM5D* significantly reduced the expression levels of *AURKB* and *ALDH1A3* in both SAS and FaDu persister cells, indicating that *KDM5D* regulates *AURKB* and *ALDH1A3* expression (Figure 4B). As such, the presence of the *KDM5D*/*AURKB* axis was then confirmed and activated in platinum-tolerant HNSCC cells. The downregulation of *ALDH1A3* in response to *KDM5D* silencing also suggested the role of the *KDM5D*/*AURKB* axis in modulating *ALDH1A3*-mediated cancer stemness, platinum tolerance, and transition towards the quiescent state of HNSCC cells. Further characterization and functional perturbation assay were then examined to demonstrate the functional role of *KDM5D* related to drug tolerance.

A key feature of cancer stemness is their self-renewability potential, which is presumably high in drug-tolerant persister cells. *KDM5D* knockdown significantly reduced tumor sphere formation in both PT-SAS and PT-FaDu cells (Figure 4C). This finding indicates that *KDM5D* contributes to the cancer stemness phenotype in cisplatin-tolerant persister cells in HNSCC. Cell cycle arrest is an essential feature of cells in the diapause state, resembling embryonic cell development. We suppressed *KDM5D* expression through short hairpin RNA (shRNA)-mediated knockdown. Both platinum-tolerant persister HNSCC cells exhibited cell cycle arrest, as indicated by a significant increase in the G0/G1 subpopulation and a decrease in the S and G2/M subpopulations (Figure 4D). The findings indicate that *KDM5D* is an essential gene that promotes cell cycle arrest and activates the diapause state in cisplatin-tolerant persister HNSCC cells. The inhibition of *KDM5D* expression resensitized platinum-tolerant persister cells upon cisplatin treatment, indicating that *KDM5D* plays a crucial role in promoting platinum tolerance in HNSCCs (Figure 4E,F). Overall, the data highlight *KDM5D*/*AURKB* axis in which *KDM5D* modulates *AURKB* expression generates platinum-tolerant persister cells, enhances cancer stemness potential, activates the diapause-like state, and alters platinum sensitivity in HNSCC.

### 2.5. KDM5D Protects DNA Damage following Platinum Treatment in Persister Cells

Platinum agents, such as cisplatin, carboplatin, and oxaliplatin, work by forming covalent binds with DNA, leading to the formation of DNA crosslinks and thus the inhibition of DNA replication, arrest of the cell cycle, and cessation of cancer cell proliferation. To determine whether *KDM5D* affects sensitivity to platinum treatment, under either the suppression or overexpression of *KDM5D*, we examined the viability, cell cycle progression, and DNA damage of platinum-tolerant HNSCC cells following cisplatin treatment. The suppression of *KDM5D* expression increased the vulnerability of both PT-SAS and PT-FaDU cells against cisplatin, as indicated by a significant reduction in the number of colonies upon cisplatin treatment (Figure 5A). Moreover, both HNSCC cell lines with *KDM5D* overexpression exhibited increased tolerance against several incremental dosages of cisplatin (Figure 5B). The results revealed that the tolerance of HNSCC cells to cisplatin treatment is mediated by *KDM5D*, and the expression of this gene increased tolerability to platinum agents. Cisplatin promotes cell cycle arrest in cancer cells. However, the increased tolerance of platinum-tolerant HNSCC cells diminishes this effect. The distribution of the cell cycle did not significantly change after cisplatin treatment in both PT-SAS and PT-FaDU cells (Figure 5C).

Consistent with the previous result presented in Figure 4D, the inhibition of *KDM5D* through shRNA-mediated knockdown induced cell cycle arrest in PT-SAS and PT-FaDU cells, as indicated by an increase in the G1 cell subpopulation and a decrease in the S cell subpopulation (Figure 5D). After cisplatin treatment, cell cycle arrest was markedly enhanced after the knockdown of *KDM5D*. Thus, *KDM5D* appears to be a key factor determining the vulnerability to platinum agents by governing the cell cycle progression of persister HNSCC cells. Platinum agents can eliminate cancer cells by forming DNA crosslinks, resulting in DNA damage. The tolerability of persister HNSCC cells to platinum agents can be mediated by lowering their susceptibility to DNA damage. To determine whether *KDM5D* alters the rate of DNA damage, we suppressed *KDM5D* expression and examined the extent of DNA damage by quantifying phosphorylation of H2A histone family member X (γH2AX), a well-known immunofluorescence marker for localizing DNA damage [38]. Following platinum-DNA adduct formation and the recruitment of DNA repair foci in response to double-strand breaks, fluorescent subnuclear foci were detected by H2AX immunofluorescence staining. Moreover, we determined the proliferation rate following cisplatin-induced DNA damage through 5-ethynyl-2’-deoxyuridine (EdU) fluorescence staining. Representative images of γH2AX and EdU staining in both PT-SAS and PT-FaDU cells were shown in Figure 5G. In the present study, *KDM5D* inhibition markedly increased the expression of γH2AX during platinum treatment while EdU expression was significantly reduced in both PT-SAS and PT-FaDu cells (Figure 5E,F). We also compare results obtained for KDMD5 and AURKB overexpression/knockdown with an established positive marker for cancer stemness (SRY-Box transcription factor 2 or SOX2), cisplatin resistance (Cyclins D1) and diapause (NRF2) as shown in Appendix A. The findings suggest a higher extent of DNA damage in response to *KDM5D* silencing, reflecting the protective role of *KDM5D* to prevent DNA damage while maintaining the proliferation of persister HNSCC cells following cisplatin treatment.

### 2.6. AURKB-Induced Mitotic Catastrophe Is Disrupted in Persister Cells

Our findings suggested that *KDM5D* promotes platinum tolerance by upregulating *AURKB* expression while maintaining cell cycle progression and preventing DNA damage. On the basis of this conjecture, we induced the aberrant activation of AURKB in platinum-tolerant HNSCC cells; *AURKB* can control the cell cycle by modulating the p53 checkpoint pathway. Therefore, by using barasertib, an AURKB inhibitor, we investigated whether AURKB affects the regulation of the cell cycle in cisplatin-tolerant persister HNSCC cells. Treatment with barasertib induced cell cycle arrest in both PT-SAS and PT-FaDU cells by markedly increasing the G1 cell subpopulation and reducing the G2 and S cell subpopulations (Figure 6A). This pharmacological perturbation mimicked the result of the transcriptomic repression of *KDM5D* (Figure 6B), suggesting that barasertib affected the downstream of the *KDM5D*/*AURKB* axis. We evaluated cell cycle progression by determining the extent of mitosis. Despite the predominance of the G1 cell subpopulation, the mitosis rate exhibited an abnormal increase following *AURKB* inhibition in a time- and dose-dependent manner, suggesting an imbalance between cell cycle arrest and aberrant mitosis. The percentage of mitotic defects following barasertib treatment increased in a dose-dependent manner (Figure 6C). The presence of aberrant mitosis, despite cell cycle arrest and abundant mitotic defects, suggests the induction of mitotic catastrophe in persister HNSCC cells. To determine any signaling perturbation during mitotic catastrophe, we observed several cell cycle checkpoints and mitotic catastrophe markers. Inhibition of *AURKB* activated the phosphorylation of several cell cycle regulators; including Checkpoint kinase 1 protein (Chk1), Cell division control protein 2 homolog (Cdc2), and Cyclin B1 (Figure 6D). AURKB inhibition resulted in higher mitotic indexes, as revealed by Giemsa staining, and mitotic defects, as revealed by fluorescence staining (Figure 6B,C). However, Cyclin B1 protein levels continued to increase (Figure 6D), suggesting that the early mitotic phase was delayed. Moreover, enhanced mitotic defects during mitosis progression in HNSCC cells were indicative of mitotic catastrophe triggered by mitosis delay. Hence, the findings indicate the high vulnerability of platinum-tolerant persister HNSCC cells to mitotic catastrophe upon inhibition of AURKB expression.

### 2.7. Cotreatment of Cisplatin and Barasertib Prolonged Tumor Suppression Potential

To determine the preclinical efficacy of AURKB inhibition in suppressing the emergence of cisplatin-tolerant persister HNSCC cells, we used an in vivo mouse tumor xenograft model and treated it with a combination of cisplatin and barasertib. The tumor suppression potential of the drug cotreatment was examined. After 4 weeks of treatment, the tumor size of each mouse group was determined. Treatment with a combination of cisplatin and barasertib significantly reduced the growth of HNSCC tumors compared with monotherapy or vehicle treatment (Figure 7A). Body weight did not significantly differ between the combination treatment group and monotherapy and vehicle treatment groups (Figure 7B). Moreover, mice in the combination group exhibited longer survival than did those in the monotherapy or vehicle group, suggesting the superior efficacy of the combination treatment in HNSCC (Figure 7C). Tissue staining revealed a significant reduction in cellular proliferation, aggressiveness, and epithelial-to-mesenchymal transition potential, as indicated by decreased levels of Ki67, vimentin, and slug, respectively (Figure 7D). The findings suggest that combining cisplatin with barasertib is beneficial for reducing tolerance to chemotherapy following platinum treatment and tumor growth in HNSCC.

## 3. Discussion

Squamous cell carcinoma is among the most common type of head and neck cancer worldwide, occurring in the oral cavity, tongue, lip, and pharynx [3]. Multidisciplinary treatments for HNSCC encompass surgery, anticancer drug therapy, and radiotherapy. Among them, anticancer drug therapy is undergoing rapid development, and various combinations of anticancer drugs are increasingly being used, including anti-EGFR and immune checkpoint inhibitors [4,5,39]. Since the advent of cisplatin in the 1970s, various cytotoxic anticancer agents have been indicated for HNSCC. However, several randomized clinical trials have demonstrated that the treatment response remains limited [4,5]. Cisplatin alone does not improve overall survival because of poor tumor response, and combining cisplatin with other agents has still not resulted in a marked breakthrough in terms of the treatment response. Relapse after cisplatin treatment in HNSCC is common and associated with poor survival and more complicated treatment requirements [6,7]. This study identified relapse or disease recurrence following cisplatin treatment and proposed a strategy to reduce treatment tolerance by eliminating platinum-tolerant persister HNSCC cells.

Evidence has indicated the emergence of DTPCs in diverse types of cancer, including HNSCC; DTPCs are responsible for the development of the final and irreversible phenotype of treatment resistance and thus disease relapse [40,41,42]. Targeting and eliminating DTPCs can prevent the development of treatment-resistant cells. However, potential clinical markers for DTPCs have yet to be determined. In this study, by employing the multilevel transcriptomic approach and analyzing bulk-level datasets and single-cell data, we determined that *KDM5D* is associated with cisplatin resistance, cancer stemness, and diapause state in persister cells and characterized the DTPC subpopulation of HNSCC in vitro. Diapause is a reversible halt in embryonic development. CSCs exploit a diapause-like state to prevent stress, specifically chemotherapy-induced apoptosis [11]. CSCs can undergo self-renewal within a tumor. Similar to normal stem cells, CSCs can become dormant when they are resistant to most anticancer treatments, including chemotherapy, and can contribute to tumor recurrence [12]. Thus, identifying a common regulator (such as *KDM5D*) of cancer stemness, a diapause-like state, and treatment resistance may help in the development of a strategy for targeting drug-tolerant persister HNSCC cells.

In patients with advanced HNSCC, treatment effectiveness is very low, as is the case in most advanced-stage cancers. Thus, novel prognostic or predictive markers as well as alternative treatment approaches are required to improve outcomes. Histone demethylases regulate biological processes, such as cell cycle control, DNA damage responses, heterochromatin formation, and pluripotency. KDMs are frequently upregulated in HNSCC and have prognostic significance. Studies have indicated that the overexpression of *LSD1*, *KDM4*, *KDM5*, or *KDM6* can predict the survival and metastasis of patients with cancer [21]. Tumor specimens from the HNSCC cohort revealed a significantly higher expression level of KDM5D protein than normal adjacent tissues. KDM5D protein expression was particularly higher in poorly differentiated tumors, implying that it regulates the pathogenesis of HNSCC and promotes the development of poorly differentiated CSCs. In addition, patients with *KDM5D*-overexpressing HNSCC had more advanced disease, lower survival, and a higher likelihood of tumor recurrence following platinum therapy. Furthermore, early relapse induced by platinum treatment was associated with aberrant KDM5D expression, suggesting that KDM5D expression contributes to the development of treatment-tolerant persister cells. Prior studies on leukemia persister cells have suggested that early relapse is caused by the delayed proliferation of persister cells. Late relapse may be caused by persister clones that survive treatment [43]. Relapse in cancer occurs when persister cells undergo proliferation after treatment cessation. By examining KDM5D expression in HNSCC tumors, the potential generation of persister cells can be monitored. Therefore, KDM5D can serve as a predictive marker for recurrence and relapse following platinum treatment in HNSCC.

A previous study reported that residual tumors comprising treatment-tolerant persister cells mimic the embryonic diapause-like state while maintaining cancer stemness properties to survive by suppressing their Myc expression and reducing their high redox burden [44]. However, in this study, *KDM5D* upregulation was identified to be a crucial mechanism for maintaining the slow cycle and diapause state of persister HNSCC cells. Our cell cycle analysis indicated that *KDM5D* prevents the deleterious effect of cell cycle arrest during the activation of the diapause-like state upon platinum treatment. Moreover, our finding demonstrates that *KDM5D* protected HNSCC persister cells from DNA double-strand damage caused by cisplatin treatment, as indicated by pronounced γH2AX expression after *KDM5D* silencing. This finding indicated that *KDM5D* might gain the DNA repair capability of platinum-tolerant persister cells. This notion was in line with a previous study that demonstrated *KDM5D* was involved in the regulation of ATR-dependent DNA damage repair of prostate cancer as shown by increasing CHK1 and cell division cycle 25C phosphatase (CDC25c) protein expression in response to *KDM5D* knockdown [30]. Similarly, our finding also noted an increase in the phosphorylation of CHK1 and CDC25c after the inhibition of AURKB by barasertib treatment, which mimicked the phenotypical response of *KDM5D* knockdown in platinum-tolerant HNSCC cells. In addition, the nonepigenetic roles of other histone demethylase members, such as *KDM5A* and *KDM5B*, are implicated in the repair of DNA, prevention of replication stress caused by hydroxyurea (HU), and development of HU-tolerant persister cancer cells [19]. Therefore, high *KDM5D* expression might play a role in the development of platinum-tolerant persister HNSCC cells. Platinum-tolerant persister HNSCC cells can be eliminated by recognizing perturbations related to the function of *KDM5D*.

Our study suggests that *KDM5D* regulates persister head and neck cancer cells by modulating *AURKB* expression. *KDM5D* is a lysine-specific demethylase that alters gene expression associated with cell cycle control and mitotic regulation by demethylating H3K4me3 and H3K4me2, which are also referred to as transcriptionally active chromatin [30]. In this case, we speculated that demethylation of H3K4me3 by KDM5D might lead to aberrant mRNA expression levels of mitotic-associated *AURKB* genes during platinum tolerance acquisition in HNSCC. As demonstrated in other studies, the demethylase activity of the KDM5 family is capable of activating cell cycle gene expression by determining H3K4me3 methylation levels at certain promoters [45]. Additionally, H3K4me3 demethylation has been shown to affect *AURKB* and *E2F2* transcription levels in breast cancer tumors and is correlated with poor clinical outcomes [24]. Disruption of *KDM5D*/*AURKB* circuitry through the use of the AURKB inhibitor barasertib induced excessive chromosomal misalignment and abnormal segregation in mitotic cells as well as promoted mitotic catastrophe in persister cells. Barasertib cotreatment enhanced the tumor suppression potential of cisplatin therapy in the HNSCC mouse xenograft model. These results are similar to those of other studies indicating that the inhibition of AURKB promoted mitotic catastrophe and markedly increased apoptotic and necrotic death as well as enhanced treatment sensitivity in cancer cells [26,46]. Aurora kinase families include serine/threonine kinases that regulate the cell cycle and thus play a major role in mitosis. Aurora kinase plays a key role in many crucial mitotic processes, including centrosome maturation, chromosome alignment, chromosome segregation, and cytokinesis [47]. Treatment-tolerant persister cells are relatively susceptible to ferroptosis induction, and this susceptibility can be used to suppress the development of DTPCs [40,41,42]. The present study indicates alternative mechanisms for exploiting persister HNSCC cells, such as targeting mitotic catastrophe.

## 4. Materials and Methods

### 4.1. Microarray and Bulk Tumor RNA Sequencing Data Acquisition

Several representative microarray datasets with data on oral squamous carcinoma cells (GSE9844) [48], head and neck cancer stem cells (GSE72384) [49], and cisplatin-resistant HNSCC cells (GSE102787) [50] were each acquired from the Gene Expression Omnibus (GEO) database portal. Moreover, the clinical information of each sample was obtained from the GEO database portal. We obtained bulk tumor RNA-sequencing data and clinical information of patients with head and neck cancer from The Cancer Genome Atlas, with access provided by the CSC Xena browser (https://xenabrowser.net/ acssesed on 23 October 2022) [51]. For each microarray dataset, normalization and probe-to-gene annotation were performed. For RNA-sequencing datasets, data normalization was previously performed by the web portal and presented in the log2 format. The overall survival data of each patient were collected for further analysis.

### 4.2. Single-Cell Profiling of HNSCC

A representative single-cell transcriptome profiling dataset by Puram et al. was selected; they explored ecosystem heterogeneity between HNSCC cells and identified the expression of genes of interest in each cell cluster. Puram et al. deposited their dataset comprising approximately 6000 cells, in the GEO database under the code GSE103322 [52]. After obtaining the file matrix, we employed the Seurat package (version 4.0.6) in R (version 4.0.1, R Core Team, Vienna, Austria) to reconstruct Seurat objects. The data were subject to generic preprocessing procedures, including the filtering of data on nonexpressed genes and the reduction in noise from data on weakly expressed mitochondrial genes. Subsequently, we performed normalization and data scaling for the Seurat objects corresponding to the data, followed by dimensionality reduction and cell cluster generation by employing the t-distributed stochastic neighbor embedding (tSNE) module. The positive and negative markers of each cluster were generated and listed. tSNE plots, dot plots, and bar graphs were constructed to determine the expression of the genes of interest between each cluster.

### 4.3. Identification of Differentially Expressed Genes

A normalized microarray dataset was prepared, and phenotypes for each sample were preset. The limma package (version 3.52.2) was used to determine the difference in the fold change and the significance level thereof for each gene. Genes with a log2 fold change of ≥1 and a *p* value of <0.05 were considered differentially expressed genes (DEGs). DEGs from each dataset were identified and presented in a Venn diagram to determine shared and common DEGs between the phenotypes of interest. Some top DEGs were selected and presented in a heatmap by using pheatmap (version 1.0.12) to determine their ability to cluster two phenotypes.

### 4.4. Pathway Enrichment Analysis

To investigate key signaling pathways and biological processes perturbed in cancer stem cells and cells in the diapause-like state in HNSCC, we performed gene set enrichment analysis (GSEA) and gene set variation analysis, respectively. Several functional annotations and curated gene sets were used, including Hallmark, KEGG, Gene Ontology, and WikiPathways. Gene sets with a *p*-value of <0.05 were selected.

### 4.5. Immunohistochemistry Staining

A total of 100 HNSCC tissue specimens from the TMU-SHH HNSCC cohort were evaluated to detect KDM5D protein expression through immunohistochemistry (IHC) staining. Formalin-fixed, paraffin-embedded tissue sections were placed on coated glass slides. IHC staining was performed using the EnVision FLEX Mini Kit (Agilent, Santa Clara, CA, USA) in accordance with the manufacturer’s protocol. Rabbit anti-KDM5D monoclonal antibody was used at 1:200 dilution. Images were captured using a microscope (Leica Microsystems, Wetzlar, Germany). In addition, images were obtained using a 100× objective lens on an Olympus BX53 microscope (Olympus, Tokyo, Japan) for scoring and data analysis. KDM5D expression in the tissue specimens was examined by calculating the IHC Q or quick score on the basis of the intensity and extent of expression. The Q-score method was applied per a previous study. The intensity (i) was quantified using the following criteria: 0 = negative, 1 = weak, 2 = medium, and 3 = high. The extent of expression was evaluated as the percentage of the entire tumor area that was positively stained. The final IHC Q score was determined by quantifying the intensity score and positivity percentage. The minimum IHC score was 0, whereas the maximum score was 300. Subsequently, we categorized tissue specimens with high and low expression of KDM5D by using the median Q-score as the cutoff for the entire cohort. The IHC Q score was calculated to determine associations among certain clinical parameters of each patient. The study was conducted in accordance with the Declaration of Helsinki, and approved by the Joint Institutional Review Board of Taipei Medical University (protocol code N202201123 and date of approval of 3 February 2022).

### 4.6. Cell Line Culture

The human HNSCC cell line FaDu was purchased from the American Type Culture Collection (Manassas, VA, USA). SAS cells were kindly provided by Prof. Michael Hsiao (Genomic Research Center, Academia Sinica, Taipei City, Taiwan). Both the cell lines were cultured in Dulbecco’s modified Eagle’s medium (DMEM, Invitrogen Life Technologies, Carlsbad, CA, USA) supplemented with 10% fetal bovine serum and 1% penicillin/streptomycin (Invitrogen) and incubated at 37 °C in a 5% humidified CO_2_ incubator. The HNSCC cells were passaged at 98% confluence, and the medium was changed every 72 h before exposure to cisplatin treatment.

### 4.7. shRNA-Mediated KDM5D Knockdown

KDM5D was knocked down in SAS and FaDu cells by using a lentiviral approach. A short hairpin RNA (shRNA) construct was purchased from Origene TR30021V (Cat#: TL309236V). Lentiviruses containing constructs targeting KDM5D or scrambled controls were generated by transfecting HEK293T cells with Lipofectamine 3000 (Thermo Fisher Scientific, Waltham, MA, USA). The viral supernatant was used to transduce SAS or FaDu cells in the presence of polybrene (4 μg/mL; Sigma-Aldrich, St Louis, MO, USA). The transduced cells were selected in a medium containing puromycin (3 μg/mL, Sigma-Aldrich, St. Louis, MO, USA), and the knockdown efficiency was determined through immunoblotting.

### 4.8. Establishment of Platinum-Tolerant Persister Cells

We generated cisplatin-tolerant persister cells by using the approach described in a previous study with some modifications. In brief, SAS and FaDu cells were treated with three cycles of 5 mg/mL cisplatin for 24 h in each cycle. The surviving cells were allowed to recover for 3 to 4 weeks before receiving the next treatment. After recovery, one cycle of cisplatin treatment was administered, and the residual viable cells were defined as platinum-tolerant persister cells. Changes in tolerance to cisplatin were indicated by the half-maximal inhibitory concentration (IC_50_) as determined using the drug-response curve for each cell line.

### 4.9. Tumorsphere Formation

The platinum-tolerant HNSCC cells (PT-SAS and PT-FaDu) were seeded in serum-free low-adhesion culture plates containing of stem cell media which the content as follows: RPMI1640 with B27 supplement (Invitrogen, Waltham, MA, USA), 20 ng/mL EGF, and 20 ng/mL basic-FGF (stem cell medium; PeproTech, Rocky Hill, NJ, USA). The cells were grown for about 14 days to allow for the formation of spheres. The spheres were then counted under a microscope and spheres formation efficiency was calculated as the ratio of the number of spheres formed to the seeded adherent cell number.

### 4.10. Immunofluorescence Staining

Immunofluorescence staining was performed in SAS and FaDu cells. Initially, the cells were plated in six-well chamber slides and then permeabilized by being treated with 0.1% Triton X-100 in 0.01 M phosphate-buffered saline (PBS, pH 7.4). The cells were washed twice in PBS containing 1% bovine serum albumin (BSA) and then stained with rabbit anti-γH2AX (dilution 1:100, cat. #7631, Cell Signaling, Danvers, MA, USA) overnight. The stained cells were washed three times, resuspended in a mounting medium, and fixated on coverslips. Subsequently, 4′,6-diamidino-2-phenylindole (DAPI) was used for nuclear staining. Photographs were captured using a Leica spectral confocal fluorescence imaging system.

### 4.11. Cell Cycle Distribution Analysis

Scrambled control cells or those with *KDM5D* knockdown were exposed to cisplatin, DMSO, or barasertib for 24 h. Subsequently, the cells were harvested, washed with PBS, and fixed with ice-cold 70% ethanol at −20 °C for 30 min. Thereafter, the cells were incubated with 5 µg/mL of RNase for 30 min at room temperature and stained with propidium iodide (5 µg/mL) for 1 h. The distribution of the cells in the G1, S, or G2 phase was determined through flow cytometry.

### 4.12. Mitotic Index and Defect Measurement

To determine differences in the mitotic index in the persister cells, they were exposed to barasertib or dimethyl sulfoxide (DMSO) for 24 and 48 h. The cells were collected and stained with 50% Wright–Giemsa solution. The mitotic index was calculated using the following formula: Mitotic index = mitotic cells (stained cells)/total cell number × 100%.

To examine the extent of mitotic defects, mitotic cells were subject to immunofluorescence staining. The SAS and FaDu cells were exposed to barasertib at 500 µM for 48 h, fixed with 1.5% formaldehyde for 10 min, and permeabilized with ice-cold methanol at room temperature for 10 min. The cells were washed twice in PBS containing 1% BSA. Fluorescence staining of tubulin and the centromeric protein kinase Aurora B was performed using rabbit antitubulin (dilution 1:100, cat. #2144, Cell Signaling, Danvers, MA, USA) and rabbit anti-Aurora B (dilution 1:200, cat. #39262, ActiveMotif, Carlsbad, CA, USA), respectively, overnight in dark. The stained cells were washed three times in 1% BSA and PBS, resuspended in a mounting medium, and fixated on coverslips. Then, DAPI was used for nuclear staining. Photographs were captured using a Leica spectral confocal fluorescence imaging system. The extent of the mitotic defect was calculated using the following formula: Mitotic defect = abnormal mitotic cells/total mitotic cell number × 100%.

### 4.13. Sodium Dodecyl Sulfate–Polyacrylamide Gel Electrophoresis and Western Blot Analysis

Total protein lysates were isolated from the cancer cell lines by using radioimmunoprecipitation lysis buffer supplemented with a protease inhibitor (1×, Cat# 78430, Thermo Fisher Scientific, Waltham, MA, USA) and a phosphatase inhibitor (0.5×, Pierce Phosphatase Inhibitor Mini Tablets, Thermo Fisher Scientific). The lysates were separated through standard sodium dodecyl sulfate–polyacrylamide gel electrophoresis by using the Protean III system (BioRad, Hercules, CA, USA) and were transferred onto a polyvinylidene fluoride membrane by using the Trans-Blot Turbo Transfer System (BioRad). Primary antibodies against phospho-Chk1 (Ser345, #2348, 1:1000), Cdc25c (Ser2, #77055, 1:1000), cyclin B1 (#4138, 1:1000), and β-actin (#3700, 1:1000) purchased from Cell Signaling Technology (Danvers, MA, USA) were used in this study. The full size blots of respective western blotting result was provided in Appendix A.

### 4.14. Tumor Xenograft Animal Study

Athymic male BALB/C nu/nu mice aged 6 weeks (n = 20, median weight = 22 ± 1.5 g) were obtained from BioLASCO (Taipei City, Taiwan) and used in in vivo experiments. In an animal center facility, the mice were maintained under regular conditions in a 12-h light/dark cycle and provided food and water ad libitum. The study procedures were approved by the Institutional Animal Care and Use Committee of Taipei Medical University and performed in an aseptic manner (LAC-2021-0671). We subcutaneously implanted 100 µL of DMEM and Matrigel mixture (Becton Dickinson, Bedford, MA, USA) containing 1 × 106 SAS cells in each mouse’s left thigh. The mice were divided into vehicle control (n = 5), cisplatin (n = 5), barasertib (n = 5), and combination (n = 5) groups. When tumors grew to approximately 100 mm3 at 10 days after the injection, either (1) cisplatin (5 mg/kg, once a week), barasertib (25 mg/kg, five times/week), or both (cisplatin + barasertib) were intraperitoneally administered in the treatment groups or (2) DMSO was administered in the control group. The dosage was determined on the basis of a previous study that evaluated the efficacy of barasertib against several types of cancer cells in vivo. The drugs were intraperitoneally administered for 4 weeks, and tumor growth was observed for 6 weeks after the initiation of treatment. We measured the tumor diameter twice per week in accordance with the institutional protocol concerning the evaluation of the effects of antitumor treatment on a subcutaneously developed human tumor xenograft model. Tumor growth was examined using a Vernier caliper, and tumor volume (V) was calculated as follows: V = 0.5 × (long diameter × short diameter^2^). The effectiveness of the treatments was evaluated on the basis of the tumor mass. After the completion of the experiment on week 6, the mice were sacrificed, and tumor masses were collected and used in subsequent experiments.

### 4.15. Statistical Analysis

A mean and a standard error of the mean were calculated for each numerical variable. Frequency and percentage were used to represent categorical variables. The mean values between the two groups were compared by an unpaired Student’s *t*-test. Analysis of variance (ANOVA) was used to identify discrepancies between groups. In cases where ANOVA results were significant, the least significant difference test was employed to test for differences between groups. In order to compare different groups under different timelines, we applied a two-way ANOVA with repeated measures. In order to determine the correlation strength between the different parameters, Pearson’s linear correlation was applied. Statistical significance was determined by a *p*-value lower than 0.05. All tests were conducted in triplicate and analyzed using R studio (version 1.4.1717, Boston, MA, USA) and GraphPad Prism (version 8.02, San Diego, CA, USA).

## 5. Conclusions

Our results indicate that barasertib effectively suppresses the expression of AURKB (Figure 8), resulting in the downregulation of *KDM5D* and enhancement of the antitumor efficacy of cisplatin by inducing the expression of apoptosis-related genes in platinum-tolerant head and neck cancer cells and thus maximizing its therapeutic potential for patients with HNSCC. In this study, we highlight the significance of KDM5D in identifying and promoting the development of platinum-tolerant persister cells in HNSCC, resulting in relapse and recurrence after platinum therapy. AURKB disruption affects the vulnerability of persister cells in a mitotic catastrophe–dependent manner, potentially overcoming platinum tolerance.

## Figures and Tables

**Figure 1 ijms-24-05310-f001:**
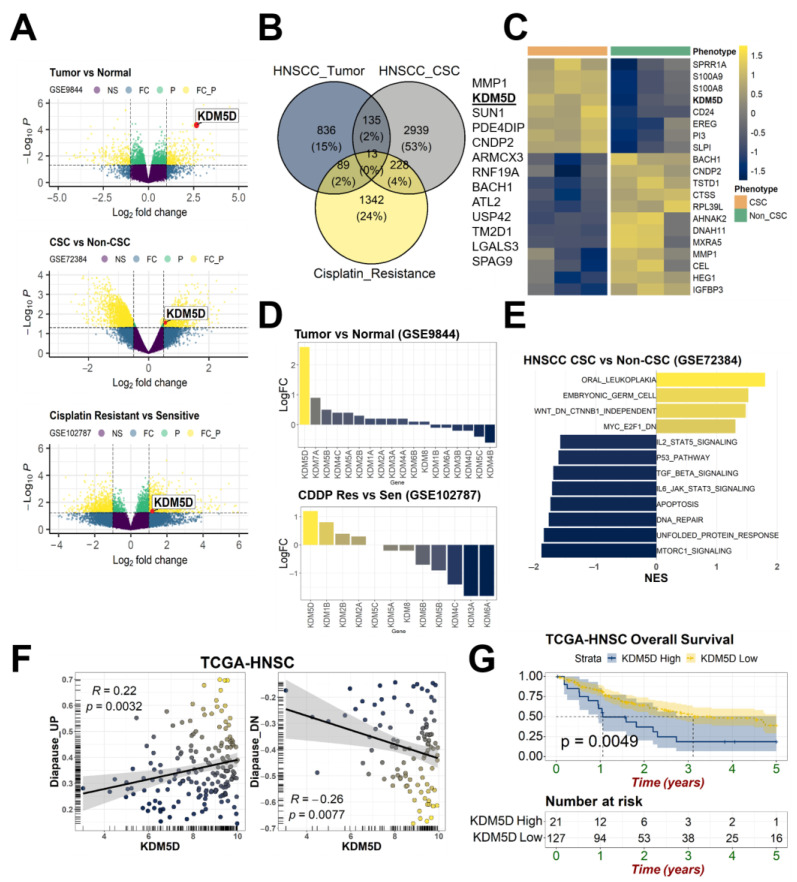
Differentially expression of *KDM5D* linked HNSCC Tumors, CSCs, and Cisplatin Resistance. (**A**) Individual volcano plots depicted overexpression of *KDM5D* as DEGs in respective phenotypes and datasets: Tumor versus Normal (GSE9844), CSCs versus Non-CSCs (GSE72384), and Cisplatin Resistant versus Sensitive (GSE102787). The position of *KDM5D* was marked in red dot with respective arrow. (**B**) Venn diagram depicts shared common DEGs between HNSCC tumors, CSCs subset, and cisplatin resistant cells, in which *KDM5D* belongs to one of the shared genes. (**C**) The heatmap of GSE72384 microarray dataset described cluster analysis of previously described top DEGs, which appropriately categorized each sample into CSCs and non-CSCs subsets. (**D**) Compared with all recognizable lysine demethylases belonging to the JARID or KDM family in humans, *KDM5D* was relatively highly expressed in HNSCC tumors and cisplatin-resistant cells. (**E**) The GSEA findings revealed that several crucial signaling pathways were activated or deactivated and linked to cancer stemness in HNSCC. (**F**) The scatter plot revealed that *KDM5D* expression was positively correlated with genes upregulated in the diapause-like state (r = 0.22, *p* = 0.0032) and negatively correlated with genes downregulated in the diapause-like state (r = −0.26, *p* = 0.0077). Diapause gene set scores of each sample were calculated by GSVA method. (**G**) A higher *KDM5D* expression level was associated with poorer overall survival in patients in the TCGA-HNSC dataset (*p* = 0.0049). NS: Not Significant, FC: Significant in Log2 Fold-Change, *p*: Significant in *p*-value, FC_P: Significant in Log2 Fold-Change and *p*-value, NES: Normalized Enrichment Score.

**Figure 2 ijms-24-05310-f002:**
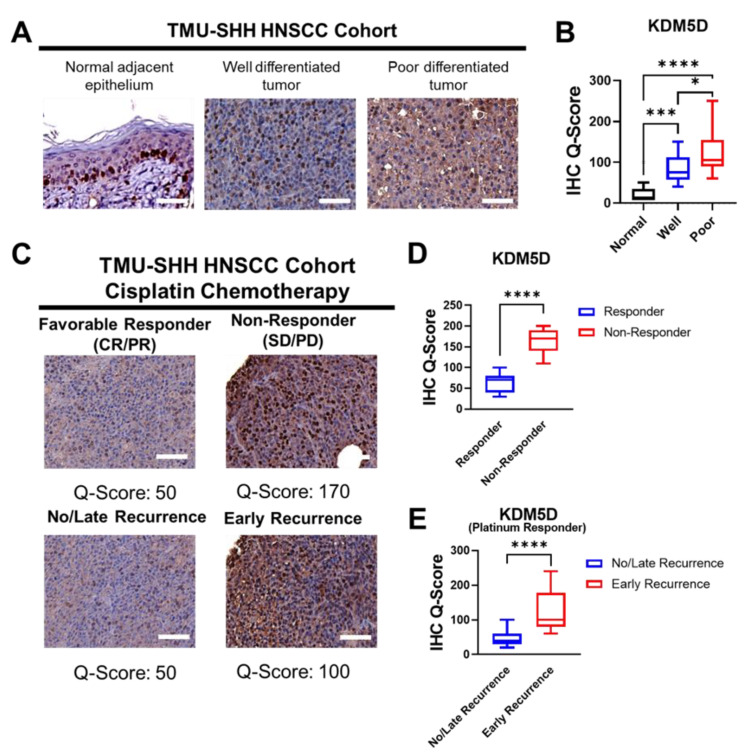
Overexpression of KDM5D was associated with poor platinum responses in HNSCC patients. (**A**) Representative images of KDM5D staining in tissue specimens of TMU-SHH HNSCC cohort in respective order: normal adjacent epithelium, well-differentiated tumor, and poorly differentiated tumors. Higher KDM5D expression was noted in HNSCC tissues than in adjacent normal epithelial tissues. (**B**) The highest KDM5D expression was observed in poorly differentiated squamous cell carcinoma tissues. (**C**) Representative images of KDM5D staining in tissue specimens of TMU-SHH HNSCC cohort according to cisplatin response and recurrence disease. (**D**) Among the patients with HNSCC, KDM5D expression was higher in the platinum non-responders than in the responders. (**E**) Among HNSCC patients who responded to platinum-based chemotherapy, higher KDM5D expression was observed in those with early disease recurrence than those with no/late-recurrence. Significance level: * *p* < 0.05; *** *p* < 0.001; **** *p* < 0.0001. Scale bar: 200 μm.

**Figure 3 ijms-24-05310-f003:**
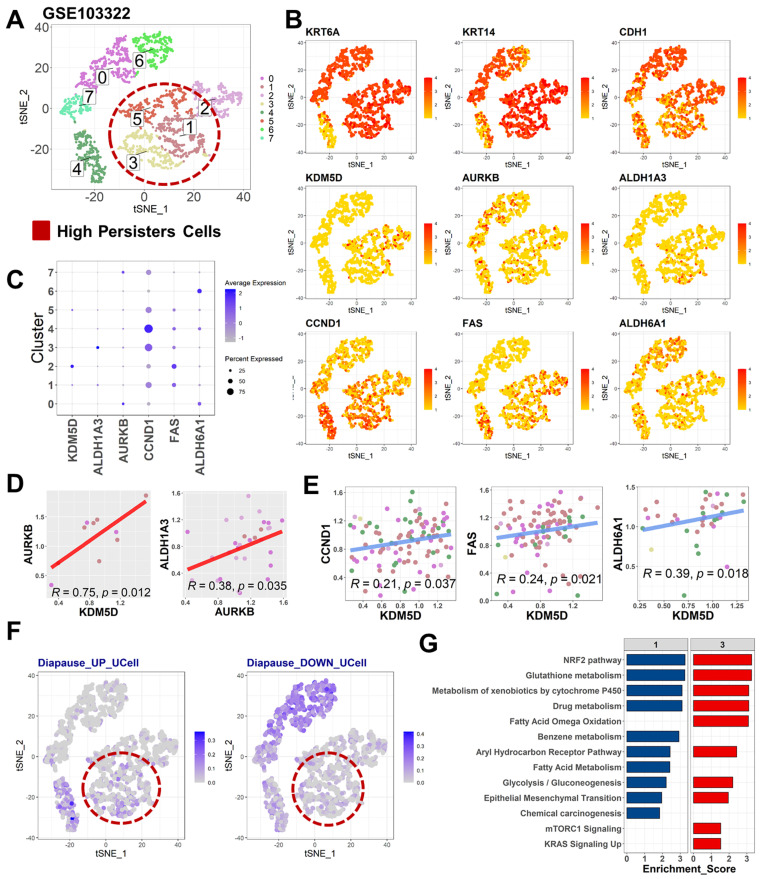
*KDM5D* and *AURKB* co-expression delineates cluster of HNSCC persister cells. (**A**) Representative tSNE plots of single-cell profiling in GSE103322 dataset showed eight distinct clusters of cells. (**B**) Array of tSNE plots portrayed expression level of interest genes such as HNSCC tumor markers (*KRT6A*, *KRT14*, *CDH1*), CSCs and persister marker (*ALDH1A3*), putative main targets of this study (*KDM5D* and *AURKB*), and diapause-related genes (*CCND1*, *FAS*, *ALDH6A1*). (**C**) Dot plot described level of expression of each gene (*KDM5D*, *ALDH1A3*, *AURKB*, *CCND1*, *FAS*, *ALDH6A1*) in eight distinct clusters. (**D**) Scatter plot depicted co-association of each interest gene; Pearson’s coefficient and *p*-value was provided in the top margin. The *KDM5D* expression was significantly correlated to *AURKB* (r = 0.75, *p* = 0.012) and *ALDH1A3* (r = 0.38, *p* = 0.035). (**E**) Scatter plot portrayed correlation between several diapause-related genes and *KDM5D*, such as *CCND1* (r = 0.21, *p* = 0.037), *FAS* (r = 0.24, *p* = 0.021), and *ALDH6A1* (r = 0.39, *p* = 0.018). (**F**) tSNE plot illustrates Diapause signature scores in each individual tumor cell. The Diapause_UP module score consisted of gene scores that are overexpressed during diapause, while Diapause_DOWN module score comprised genes that are downregulated at diapause stage. Persister cells were enriched among the clusters within the circle marker (red dash line). Predominant *ALDH1A3* and *KDM5D* expression were noted in cell clusters no. 1, 3, and 5 which was consistent with diapause state activation. (**G**) The clusters, which were speculated to activate diapause state (clusters no. 1 and 3) exhibited several common features of diapause state, including the activation of NRF2, glutathione, drug metabolism, glycolysis, and epithelial-mesenchymal transition pathways.

**Figure 4 ijms-24-05310-f004:**
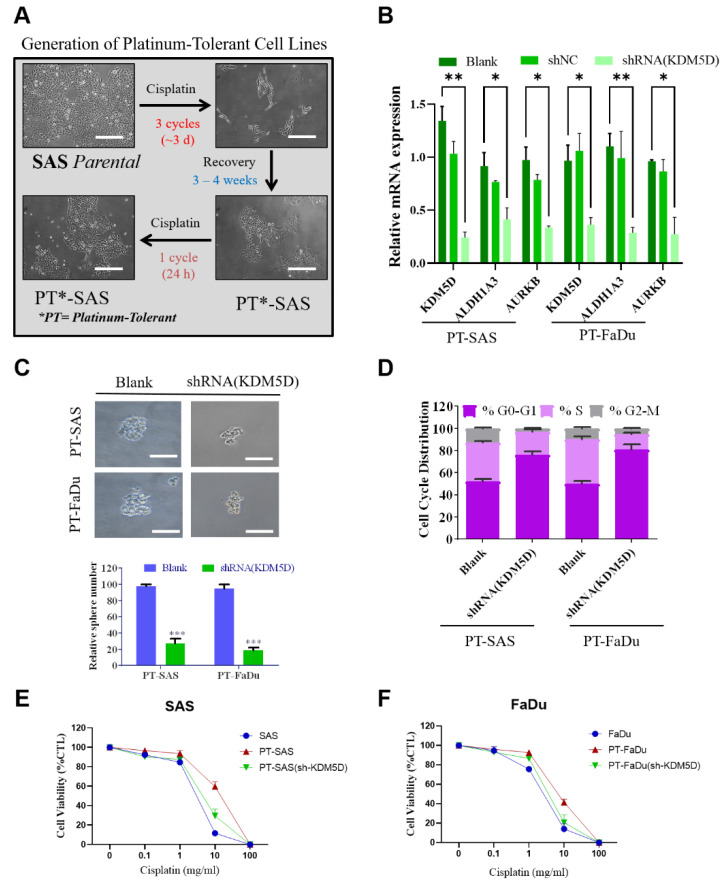
*KDM5D* promotes the generation of platinum-tolerant persister cells. (**A**) Brief schematic figure shows the steps to generate cisplatin-tolerant persister cells. Parental HNSCC cells were treated with a short-course of cisplatin treatment followed by a ‘drug-holiday’ or recovery stage and a final cisplatin course. At the end of this stage, the HNSCC cells were relatively viable and exhibited increased platinum tolerance. (**B**) *KDM5D* knockdown significantly reduced the expression levels of *AURKB* and *ALDH1A3* in both SAS and FaDu persister cells, indicating that *KDM5D* regulates *AURKB* and *ALDH1A3* expression. (**C**) *KDM5D* knockdown significantly reduced tumor sphere formation in both PT-SAS and PT-FaDu cells. (**D**) Both platinum-tolerant persister HNSCC cells exhibited cell cycle arrest, as indicated by a significant increase in the G0/G1 subpopulation and a decrease in S and G2/M subpopulations. (**E**,**F**) *KDM5D* silencing re-sensitized platinum-tolerant persister cells upon cisplatin treatment, indicating that *KDM5D* plays a crucial role in promoting platinum tolerance in HNSCCs. Significance level: * *p* < 0.05; ** *p* < 0.01, *** *p* < 0.001. Scale bar: 100 μm.

**Figure 5 ijms-24-05310-f005:**
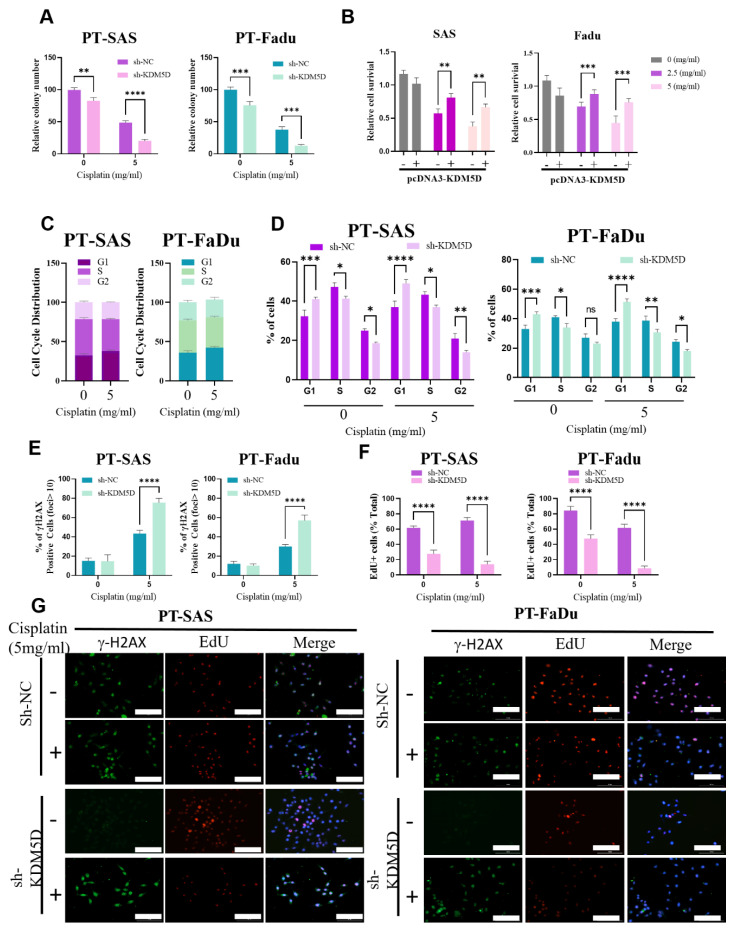
*KDM5D* abrogates DNA damage and cell cycle arrest upon platinum treatment. (**A**) The suppression of *KDM5D* expression increased the vulnerability of both PT-SAS and PT-FaDU cells against cisplatin, as indicated by a significant reduction in the number of colonies upon cisplatin treatment. (**B**) Both HNSCC cell lines (SAS and FaDu) with *KDM5D* overexpression exhibited increased tolerance against several incremental dosages of cisplatin. (**C**) The distribution of the cell cycle did not significantly change after cisplatin treatment in both PT-SAS and PT-FaDU cells. (**D**) *KDM5D* silencing through shRNA-mediated knockdown induced cell cycle arrest in PT-SAS and PT-FaDU cells, as indicated by an increase in the G1 cell subpopulation and a decrease in the S cell subpopulation. (**E**) The level of DNA damage upon cisplatin treatment was significantly increased in response to *KDM5D* silencing, as indicated by the percentage of γH2AX positive cells with >10 foci in both PT-SAS and PT-FaDu cells. (**F**) Cellular proliferation was significantly reduced following *KDM5D* silencing and become more suppressed after cisplatin treatment, as reflected by decrease in EdU-positive cell fraction in both PT-SAS and PT-FaDu cells. (**G**) Representative immunofluorescence images describe differential expression of DNA damage marker, γH2AX and proliferation marker, EdU following *KDM5D* knock-down and cisplatin treatment. Silencing of *KDM5D* markedly increased the expression of γH2AX but reduced that of EdU in both PT-SAS and PT-FaDu cells, suggesting a protective role of *KDM5D* to attenuate cisplatin-mediated DNA damage. Significance level: * *p* < 0.05; ** *p* < 0.01; *** *p* < 0.001; **** *p* < 0.0001. Scale bar: 100 μm.

**Figure 6 ijms-24-05310-f006:**
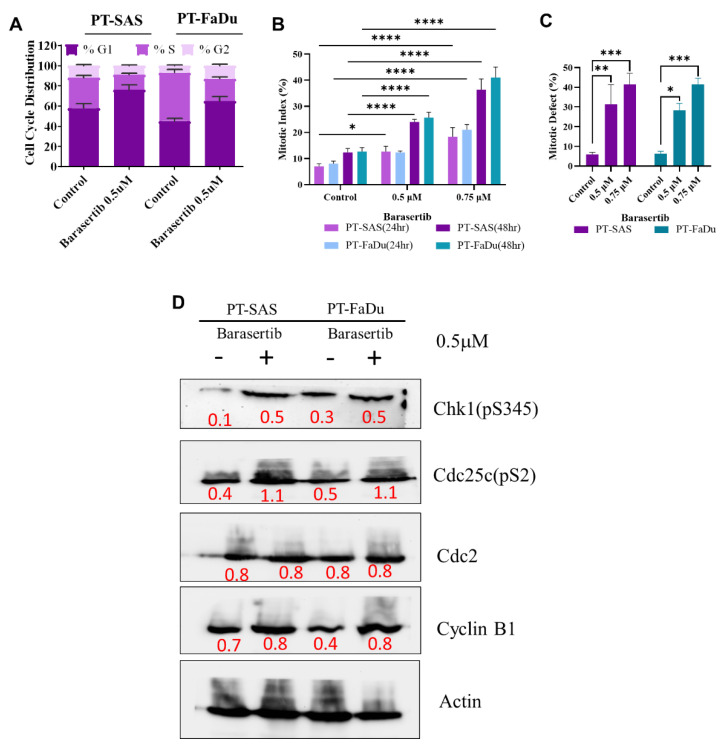
AURKB inhibition promoted mitotic catastrophe in HNSCC persister cells. (**A**) Treatment with AURKB inhibitor, barasertib, induced cell cycle arrest in both PT-SAS and PT-FaDU cells by markedly increasing the G1 cell subpopulation while reducing the G2 and S cell subpopulations. (**B**) Bar graph described dose-dependent and time-dependent significant perturbation of mitotic activity of PT-SAS and PT-FaDU cells upon barasertib treatment. (**C**) The percentage of mitotic defects following barasertib treatment increased in a dose-dependent manner. (**D**) AURKB inhibition modulated several cell cycle regulators and activated mitotic catastrophes markers, such as Chk1, Cdc25 and Cyclin B1, suggesting the inhibition of cell cycle progression despite the enhancement of mitosis. Significance level: * *p* < 0.05; ** *p* < 0.01; *** *p* < 0.001; **** *p* < 0.0001.

**Figure 7 ijms-24-05310-f007:**
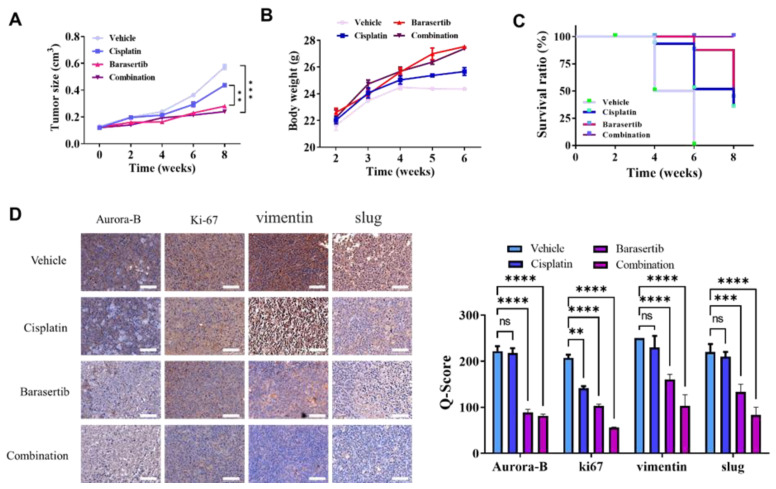
Cisplatin and barasertib co-treatment extend tumor suppression potential in vivo. (**A**) Treatment with a combination of cisplatin and barasertib significantly reduced the growth of HNSCC tumors compared with monotherapy or vehicle treatment. (**B**) Body weight did not significantly differ between the combination treatment, monotherapy, and vehicle treatment groups. (**C**) Mice in the combination group exhibited longer survival than did those in the monotherapy or vehicle group, suggesting the superior efficacy of the combination treatment in HNSCC. (**D**) Tissue staining revealed a significant reduction in cellular proliferation, aggressiveness, and epithelial-to-mesenchymal transition potential, as indicated by decreased levels of Ki67, Vimentin, and slug, respectively. Significance level: ns: not significant; ** *p* < 0.01; *** *p* < 0.001; **** *p* < 0.0001. Scale bar: 200 μm.

**Figure 8 ijms-24-05310-f008:**
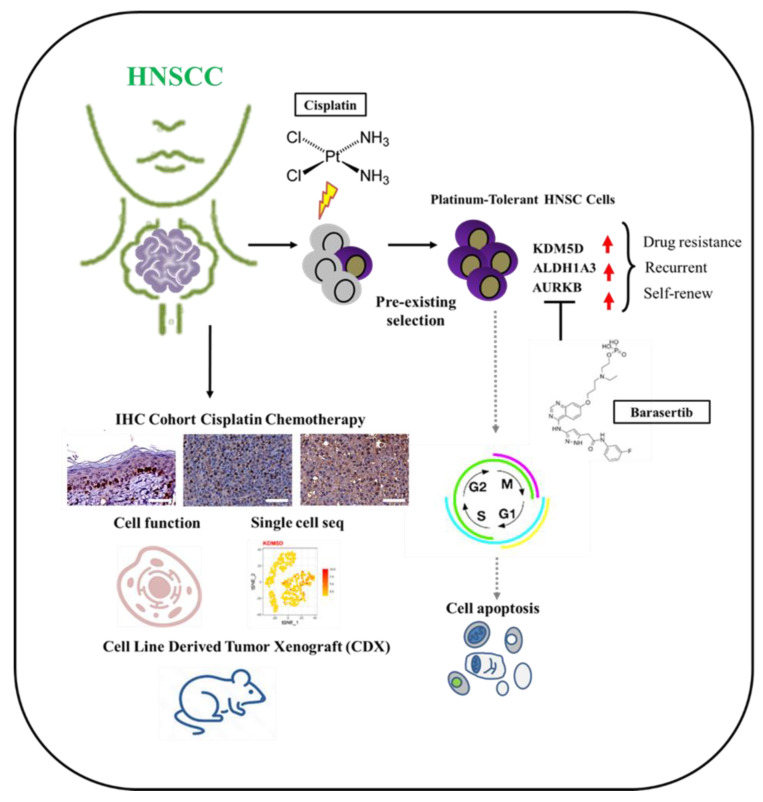
Schematic illustration of *KDM5D* contribution to affect the clinical outcome and biological development of platinum-tolerant persister cells in HNSCC. Left panel schema shows association between high KDM5D expression and poor clinical outcome in HNSCC patients encompassing poor response to platinum treatment or early recurrence disease. Right panel schema illustrates development of platinum-tolerant persister cells characterized by high expression of *KDM5D*/*AURKB* axis which disrupts AURKB by barasertib treatment deregulated tolerance mechanism and promoted mitotic catastrophe in platinum-tolerant cells.

**Table 1 ijms-24-05310-t001:** Clinicopathological Association between KDM5D Expression and Treatment Outcome of TMU-SHH HNSCC Patients Cohort.

Clinical Outcome	KDM5D		χ^2^	*p*
	High	Low		
Total Cohort (*n* = 100)				
No Responses (SD/PD) *	21 (70.0%)	9 (30.0%)	7.83	0.004
Favorable Responses (CR/PR) *	26 (37.1%)	44 (62.9%)		
Platinum Responder cohort (*n* = 70)				
Early Recurrence (<12 months)	12 (60.0%)	8 (40.0%)	6.26	0.012
No/Late Recurrence (>12 months)	14 (28.0%)	36 (72.0%)		

* Treatment response was assessed as per RECIST criteria.

## Data Availability

The Datasets that are used and analyzed by the current investigation will be provided by the corresponding author in reply to the reasonable demands. Experimental procedures, characterization of new compounds, and all other data supporting the findings are available in the Appendix A.

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
