# Peer review of "KDM5D Histone Demethylase Identifies Platinum-Tolerant Head and Neck Cancer Cells Vulnerable to Mitotic Catastrophe"

_ijms, 2023, doi:10.3390/ijms24065310_

Round 1

Reviewer 1 Report

In the manuscript entitled “KDM5D Histone Demetyhlase Identifies Platinum-Tolerant Head and Neck Cancer Cells Vulnerable to Mitotic Catastrophe,” submitted by Chen et al., molecular markers of the cisplatin-tolerant persister cells were searched and studied. Local recurrence of Head and neck squamous cell carcinoma (HNSCC) after surgical resection and ensuing chemoradiotherapy has been thought to be the major cause of mortality. In this recurrence, cisplatin-tolerant persister cells occur, and conventional anticancer therapies did not successfully remove those persister cells. Therefore, the authors searched for and studied the genes that can make the HNSCC-derived cisplatin-tolerant persister cells sensitive to the anti-cancer platinum treatment. The authors found that a histone demethylase KDM5D is overexpressed in the cisplatin-tolerant persister cells, and that high expression levels of KDM5D are associated with poor responses to anti-cancer treatment and with early recurrence of HNSCC. In contrast, knockdown of KDM5D expression resulted in the regained cisplatin sensitivity. Besides, by bioinformatic and experimental analyses, the authors found that KDM5D regulates mRNA levels of the Aurora kinase B (AURKB) and an aldehyde dehydrogenase ALDH1A3. The authors particularly focused on AURKB, as AURKB, a part of chromosome passenger complex, is required for establishment of metaphase chromosome alignment as well as successful cytokinesis. Inhibition of AURKB by barasertib increased efficacy of the cisplatin treatment in a mouse tumor model. The authors conclude that KDM5D is a potential marker protein of the cisplatin-tolerant persister cells in HNSCC, and that co-treatment of the AURKB inhibitor and cisplatin might overcome platinum tolerance.

  As the major finding is biologically and clinically important, and as the manuscript well describes the results, I think that the manuscript is suitable for publication in International Journal of Molecular Science, after the following points are responded.

(1) line-36. What does “the KDM5D/AURKB axis” mean? This word might come from Fig 4B showing that KDM5D regulates mRNA expression levels of AURKB, but this word is unclear in the current text. If not previously reported, KDM5D regulation of the AURKB expression should be really emphasized and clearly mentioned in the Abstract.

(2) lines-88-89. “aberrant H3K4me3 active transcription” is unclear.

(3) line-105. While “DEGs” are explained in line 555, please explain the abbreviation in the first appearance.

(4) Figure 1A. The position of KDM5D is unclear in the plots. Can KDM5D be highlighted by a larger red dot or an arrow?

(5) Figure 1B. In the Venn diagram, please type the numbers larger for visibility.

(6) Figure 1G. “Numberat risk” is Number at risk?

(7) Figure 3A, 3B. Is there a good reason to write the code and gene names (GSE103322, KRT6A, etc) in red?

(8) Figure 4A. Cells are not very visible. Consider contrast and/or size of the images for better visibility.

(9) Figure 4B, and part 2.4. This is one of the most important results of the present study, but not much emphasized in the present manuscript. This result should be more emphasized in the corresponding section (2.4), and how this expression regulation is achieved should be discussed in the Discussion section.

(10) line-347. Rather than describing “a well-known marker,” the authors should describe what is gamma H2AX by citing some references.

(11) line-351. The present manuscript does not seem to discuss how “KDM5D prevents DNA damage.” This intriguing result should be discussed in the Discussion section (-> point (21)).

(12) Figure 5B. “pcDNA3” should be pcDNA3-KDM5D, if I understand this figure correctly.

(13) Figure 5E. While the message of the figure is clear, cells are too small to identify. I recommend to enlarge each panel.

(14) line-395. Please clearly explain “the enhancement of mitosis.” (e.g., an increased mitotic index measured by the Giemsa staining)

(15) Figure 6B, 6C. 0.5um, 0.75um -> 0.5 μM, 0.75 μM, if these mean concentration of barasertib.

(16) Figure 6B, 6C, 6D. The increased mitotic index and mitotic defects after AURKB inhibition (6B, 6C) and the sustained Cyclin B1 protein levels (6D) indicate that the mitotic cells caused by AURKB inhibition are delayed in early mitotic phases (prometaphase or metaphase), failing to enter anaphase. Such a description had better be added in the result.

(17) line-459. “However, potential … to be determined.” This sentence should be described in the Introduction, as this makes a major motivation of the present study.

(18) line-464. “Diapause is a …development.” This sentence should be described in the Introduction, where the word “diapause” first appears.

(19) line-473 to 475. “Histone demethylases … and pluripotency.” This basic information should be described in the Introduction.

(20) line-485 to 486. “suggesting that KDM5D … persister cells.” In the Discussion, discussions about possible molecular mechanisms how KDM5D provides cells of the treatment-tolerance have to be added, based on its molecular characteristics as a histone demethylase and the published knowledge on KDM proteins.

(21) line-499. “as indicated by decreased gH2AX expression.” This interesting point should be more discussed, for example, by explicitly asking whether KDM5D makes the HNSCC persister cells insensitive to DNA damage (DNA damage exists but cells poorly sense it), invulnerable to DNA damaging (DNA damage does not occur in persister cells), or gain the increased DNA repair capability (DNA damage occurs but is more quickly repaired in persister cells than in normal cells).

(22) line-518 to 520. Aurora kinases and their significance in cell division had better be described in the Introduction, as the present manuscript mentions Aurora kinases in the Abstract but not in the Introduction.

(23) line 522. “This study” had better be “The present study,” to avoid confusion.

Author Response

Dear Reviewer,

Coauthors and I very much appreciated the encouraging, critical and constructive comments on this manuscript by the reviewer. The comments have been very thorough and useful in improving the manuscript. We strongly believe that the comments and suggestions have increased the scientific value of the revised manuscript by many folds. We have taken them fully into account in revision. We are submitting the corrected manuscript with the suggestion incorporated in the manuscript. The manuscript has been revised as per the comments given by the reviewer, and our responses to all the comments are as follows:

Response to Reviewers:

Reviewer #1: In the manuscript entitled “KDM5D Histone Demetyhlase Identifies Platinum-Tolerant Head and Neck Cancer Cells Vulnerable to Mitotic Catastrophe,” submitted by Chen et al., molecular markers of the cisplatin-tolerant persister cells were searched and studied. Local recurrence of Head and neck squamous cell carcinoma (HNSCC) after surgical resection and ensuing chemoradiotherapy has been thought to be the major cause of mortality. In this recurrence, cisplatin-tolerant persister cells occur, and conventional anticancer therapies did not successfully remove those persister cells. Therefore, the authors searched for and studied the genes that can make the HNSCC-derived cisplatin-tolerant persister cells sensitive to the anti-cancer platinum treatment. The authors found that a histone demethylase KDM5D is overexpressed in the cisplatin-tolerant persister cells, and that high expression levels of KDM5D are associated with poor responses to anti-cancer treatment and with early recurrence of HNSCC. In contrast, knockdown of KDM5D expression resulted in the regained cisplatin sensitivity. Besides, by bioinformatic and experimental analyses, the authors found that KDM5D regulates mRNA levels of the Aurora kinase B (AURKB) and an aldehyde dehydrogenase ALDH1A3. The authors particularly focused on AURKB, as AURKB, a part of chromosome passenger complex, is required for establishment of metaphase chromosome alignment as well as successful cytokinesis. Inhibition of AURKB by barasertib increased efficacy of the cisplatin treatment in a mouse tumor model. The authors conclude that KDM5D is a potential marker protein of the cisplatin-tolerant persister cells in HNSCC, and that co-treatment of the AURKB inhibitor and cisplatin might overcome platinum tolerance.

As the major finding is biologically and clinically important, and as the manuscript well describes the results, I think that the manuscript is suitable for publication in International Journal of Molecular Science, after the following points are responded.

  1. line-36. What does “the KDM5D/AURKB axis” mean? This word might come from Fig 4B showing that KDM5D regulates mRNA expression levels of AURKB, but this word is unclear in the current text. If not previously reported, KDM5D regulation of the AURKB expression should be really emphasized and clearly mentioned in the Abstract.

Answer: It was a pleasure to receive the reviewer's thoughtful comment. According to this comment and our best knowledge, this study is the pioneering report, which suggests that KDM5D modulates the expression of AURKB and eventually contributes to the platinum tolerance of head and neck cancer. As per the suggestion, we then added a brief summary of this result to the abstract section to make it more meaningful and highlight the result. Therefore, please kindly refer to our abstract.

Updated Abstract Section, please see page 1, line 26-46.

Head and neck squamous cell carcinoma (HNSCC) is a major contributor to cancer in-cidence globally and is currently managed by surgical resection followed by adjuvant chemoradi-otherapy. However, local recurrence is the major cause of mortality, indicating the emergence of drug-tolerant persister cells. A specific histone demethylase, KDM5D, is overexpressed in diverse types of cancers and involved in cancer cell cycle regulation. However, the role of KDM5D in the development of cisplatin-tolerant persister cells remains unexplored. Here, we demonstrated that KDM5D contributes to the development of persister cells. Aurora Kinase B (AURKB) disruption affected the vulnerability of persister cells in a mitotic catastrophe–dependent manner. Comprehensive in silico, in vitro, and in vivo experiments were performed. KDM5D expression was upregulated in HNSCC tumor cells, cancer stem cells, and cisplatin-resistant cells with biologically distinct signaling alterations. In an HNSCC cohort, high KDM5D expression was associated with a poor response to platinum treatment and early disease recurrence. KDM5D knockdown reduced the tolerance of persister cells to platinum agents and caused marked cell cycle deregulation, including the loss of DNA damage prevention, and abnormal mitosis enhanced cell cycle arrest. By modulating mRNA levels of AURKB, KDM5D promoted generation of platinum-tolerant persister cells in vitro, leading to the identification of KDM5D/AURKB axis, which regulates cancer stem-ness and drug tolerance of HNSCC. Treatment with an AURKB inhibitor, namely barasertib, re-sulted in a lethal consequence of mitotic catastrophe in HNSCC persister cells. The cotreatment of cisplatin and barasertib suppressed tumor growth in the tumor mouse model. Thus, KDM5D might be involved in the development of persister cells, and AURKB disruption can overcome tolerance to platinum treatment in HNSCC.

  1. lines-88-89. “aberrant H3K4me3 active transcription” is unclear.

Answer: We thank the editor for their astute observation. Indeed, the exact meaning of this sentence is somewhat unclear, which we actually intend to describe that KDM5D deficiency results in increase of H3K4me3 methylation. Therefore, we then revised the sentence and wording into a more understandable text. Please kindly refer to our Introduction section.

Updated Introduction Section, please see page 3, line 113-115.

KDM5D deficiency results in increase of H3K4me3 methylation, leading to DNA replication stress and genomic instability [22]. This alteration increases the level of the G2/M checkpoint regulator and encourages the activation of the ATR-dependent mechanism through DNA replication stress.

  1. line-105. While “DEGs” are explained in line 555, please explain the abbreviation in the first appearance.

Answer: We thank the editor for their suggestion. Thus, based on this suggestion we then provided the explanation of this abbreviation at the result section where it firstly introduced in the main text. Please kindly refer to our Result section.

Updated Result Section, please see page 3, line 135-137.

The Differentially Expressed Genes (DEGs) are highlighted in the volcano plot of each dataset. KDM5D was upregulated in HNSCC tumors, CSCs, and cisplatin-resistant cells in HNSCC (Figure 1A).

  1. Figure 1A. The position of KDM5D is unclear in the plots. Can KDM5D be highlighted by a larger red dot or an arrow?

Answer: We thank the editor for their very constructive suggestion. According to this advice, we then reconstructed the plot and highlighted the position where KDM5D was found in red dot and an arrow. Therefore, please kindly refer to our new figure 1A.

Updated figure 1A, please see page 5.

Updated legend of figure 1A, please see page 5, line 179-183.

Figure 1. Differentially expression of KDM5D linked HNSCC Tumors, CSC, and Cisplatin Resistance. (A) Individual volcano plot depicted overexpression of KDM5D as DEGs in respective phenotypes and datasets: Tumor vs Normal (GSE9844), CSC vs Non-CSC (GSE72384), and Cisplatin Resistant vs Sensitive (GSE102787). The position of KDM5D was marked in red dot with respective arrow.

  1. Figure 1B. In the Venn diagram, please type the numbers larger for visibility.

Answer: We thank the editor for their very constructive and positive advice. According to this advice, we then enlarge the number of associated genes and respective percentages inside venn diagram and gain the readability of this figure. Therefore, please kindly refer to our new figure 1B.

Updated figure 1B, please see page 5.

  1. Figure 1G. “Numberat risk” is Number at risk?

Answer: We thank the editor for their very positive advice. According to this advice, we then re-type the misspelled information into “Number at risk”. Therefore, please kindly refer to our new figure 1G.

Updated figure 1G, please see page 5.

  1. Figure 3A, 3B. Is there a good reason to write the code and gene names (GSE103322, KRT6A, etc) in red?

Answer: We thank the editor for pointing out this sharp observation. To be honest, there is no specific or rational reason to highlight the dataset identifier and gene names in red color. To minimize any confusion, we re-constructed the text into default format as black color. Therefore, please kindly refer to our new figure 3A and 3B.

Updated figure 3A and 3B, please see page 9.

  1. Figure 4A. Cells are not very visible. Consider contrast and/or size of the images for better visibility.

Answer: We thank the editor for their constructive suggestion. According to this advice, we then retouched the images into a better contrast to gain visibility. Therefore, please kindly refer to our new figure 4A.

Updated figure 4A, please see page 11.

  1. Figure 4B, and part 2.4. This is one of the most important results of the present study, but not much emphasized in the present manuscript. This result should be more emphasized in the corresponding section (2.4), and how this expression regulation is achieved should be discussed in the Discussion section.

Answer: We thank the editor for their constructive advice. Indeed, the regulation of KDM5D/AURKB axis is one of the key result of this present study and need to be highlighted in the main text. Several sentences were then added and reconstructed to convey the significance of this key result in the main text. Therefore, please kindly refer to our result and discussion section.

Updated Result Section 2.4, please see page 9-10, line 320-357.

2.4. KDM5D promoted persister cell development by upregulating AURKB

To determine the properties of platinum-tolerant persister cells in HNSCC, we generated an in vitro HNSCC cell line model. As presented in Figure 4A, after short-term cisplatin treatment followed by a period of no drug exposure, the HNSCC cells were relatively viable and exhibited increased tolerance (Figure 4A). The plati-num-tolerant cell lines were then used as basic model for dissecting the basic molecu-lar mechanism of drug tolerance acquisition. The quantitative polymerase chain reac-tion revealed relatively high expression levels of KDM5D, ALDH1A3, and AURKB in wild-type platinum-tolerant SAS (PT-SAS) and FaDu (PT-FaDU) cells (Figure 4B). According to previous in silico finding through multiple sets of transcriptomic profiling and our speculation regarding the putative role of KDM5D/AURKB axis in contributing drug-tolerant persister cells, then KDM5D silencing was performed in platinum-tolerant HNSCC cells and determine the downstream modulation by KDM5D. As expected, knockdown of KDM5D significantly reduced the expression levels of AURKB and ALDH1A3 in both SAS and FaDu persister cells, indicating that KDM5D regulates AURKB and ALDH1A3 expression (Figure 4B). As such, the presence of KDM5D/AURKB axis was then confirmed and activated in platinum-tolerant HNSCC cells. Further characterization and functional perturbation assay were then examined to demonstrate functional role of KDM5D related to drug tolerance.

A key feature of cancer stemness is their self-renewability potential, which is pre-sumably high in drug-tolerant persister cells. KDM5D knockdown significantly re-duced tumor sphere formation in both PT-SAS and PT-FaDu cells (Figure 4C). This finding indicates that KDM5D contributes to the cancer stemness phenotype in cispla-tin-tolerant persister cells in HNSCC. Cell cycle arrest is an essential feature of cells in the diapause state, resembling embryonic cell development. We suppressed KDM5D expression through shRNA-mediated knockdown. Both platinum-tolerant persister HNSCC cells exhibited cell cycle arrest, as indicated by a significant increase in the G0/G1 subpopulation and a decrease in S and G2/M subpopulations (Figure 4D). The findings indicate that KDM5D is an essential gene that promotes cell cycle arrest and activates the diapause state in cisplatin-tolerant persister HNSCC cells. The inhibition of KDM5D expression resensitized platinum-tolerant persister cells upon cisplatin treatment, indicating that KDM5D plays a crucial role in promoting platinum toler-ance in HNSCCs (Figure 4E and F). Overall, the data highlight the regulatory mechanism by which KDM5D upregulates AURKB, forming an axis between KDM5D and AURKB that generates platinum-tolerant persister cells, enhances cancer stemness potential, activates the diapause-like state, and modulates platinum sensitivity in HNSCC.

Updated Discussion Section, please see page 18, line 575-585.

Our study suggests that KDM5D regulates persister head and neck cancer cells by upregulating AURKB. KDM5D is a lysine-specific demethylase that alters gene expres-sion associated with cell cycle control and mitotic regulation by demethylating H3K4me3 and H3K4me2, which are also referred to as transcriptionally active chro-matin [22]. In this case, we speculated that demethylation of H3K4me3 by KDM5D might lead to aberrant mRNA expression levels of mitotic-associated AURKB genes during platinum tolerance acquisition in HNSCC. As demonstrated in other studies, the demethylase activity of the KDM5 family is capable of activating cell cycle gene expression by determining H3K4me3 methylation levels at certain promoters [33]. Ad-ditionally, H3K4me3 demethylation has been shown to affect AURKB and E2F2 tran-scription levels in breast cancer tumors and is correlated with poor clinical outcomes [34].

  1. line-347. Rather than describing “a well-known marker,” the authors should describe what is gamma H2AX by citing some references.

Answer: We thank the editor for their positive advice. According to this comment, we then citing several references and briefly introduced H2AX as DNA double-strand break marker. Therefore, please kindly refer to our result section.  

Updated Result Section, please see page 12 , line 399-404.

To determine whether KDM5D alters the rate of DNA damage, we suppressed KDM5D expression and examined the extent of DNA damage by quantifying γH2AX, a well-known immunofluorescence marker for localizing DNA damage [38]. Following platinum-DNA adduct formation and the recruitment of DNA repair foci in response to double-strand breaks, fluorescent subnuclear foci were detected by H2AX immuno-fluorescence staining.

Additional Reference:

Owiti, N. A.; Nagel, Z. D.; Engelward, B. P., Fluorescence Sheds Light on DNA Damage, DNA Repair, and Mutations. Trends in Cancer 2021, 7, (3), 240-248.

  1. line-351. The present manuscript does not seem to discuss how “KDM5D prevents DNA damage.” This intriguing result should be discussed in the Discussion section (-> point (21)).

Answer: We thank the editor for their critical suggestion. According to this comment, we then provided several discussion about how KDM5D might involve in protecting platinum-mediated DNA damage. Therefore, please kindly refer to our discussion section

Updated Discussion Section, please see page 17, line 559-569.

Moreover, our finding demonstrates that KDM5D protected HNSCC persister cells from DNA double-strand damage caused by cisplatin treatment, as indicated by pro-nounced γH2AX expression after KDM5D silencing. This finding indicated that KDM5D might gain the DNA repair capability of platinum-tolerant persister cells. This notion was in line with previous study that demonstrated KDM5D was involved in regulation of ATR-dependent DNA damage repair of prostate cancer as shown by in-creasing of CHK1 and CDC25c protein in response to KDM5D knockdown [22]. Similarly, our finding also noted increase phosphorylation of CHK1 and CDC25c after inhibition of AURKB by barasertib treatment, which mimicked the phenotypical response of KDM5D knockdown in platinum-tolerant HNSCC cells.

  1. Figure 5B. “pcDNA3” should be pcDNA3-KDM5D, if I understand this figure correctly.

Answer: We thank the editor for their very positive advice. Indeed, the assumption of this type of plasmid-mediated KDM5D overexpression is correct. According to this, we then re-type the misspelled information into “pcDNA3-KDM5D”. Therefore, please kindly refer to our new figure 5B.  

Updated Figure 5B, please see page 13.

  1. Figure 5E. While the message of the figure is clear, cells are too small to identify. I recommend to enlarge each panel.

Answer: We thank the editor for their very constructive advice. According to this comment, we then replaced the figure into more visible ones. Therefore, please kindly refer to figure 5E

Updated Figure 5E, please see page 13.

  1. line-395. Please clearly explain “the enhancement of mitosis.” (e.g., an increased mitotic index measured by the Giemsa staining)

Answer: We thank the editor for their very constructive advice. According to this comment, we then explained this sentence into more understandable context and meaning. Therefore, please kindly refer to result section

Updated Result Section, please see page 14, line 451-459.

Inhibition of AURKB activated the phosphorylation of several cell cycle regulators, in-cluding Chk1, Cdc2, and Cyclin B1 (Figure 6D). AURKB inhibition resulted in higher mitotic indexes, as revealed by Giemsa staining, and mitotic defects, as revealed by fluorescence staining (Figure 6B, 6C). However, Cyclin B1 protein levels continued to increase (Figure 6D), suggesting that the early mitotic phase was delayed. Moreover, enhanced mitotic defects during mitosis progression in HNSCC cells were indicative of mitotic catastrophe triggered by mitosis delay. Hence, the findings indicate the high vulnerability of platinum-tolerant persister HNSCC cells to mitotic catastrophe upon inhibition of AURKB expression.

  1. Figure 6B, 6C. 0.5um, 0.75um -> 0.5 μM, 0.75 μM, if these mean concentration of barasertib.

Answer: We thank the editor for their very positive advice. According to this, we then re-type the misspelled symbol of micro molar into more appropriate one (μM). Therefore, please kindly refer to our new figure 6B and 6C

Updated figure 6B and 6C, please see page 14.

  1. Figure 6B, 6C, 6D. The increased mitotic index and mitotic defects after AURKB inhibition (6B, 6C) and the sustained Cyclin B1 protein levels (6D) indicate that the mitotic cells caused by AURKB inhibition are delayed in early mitotic phases (prometaphase or metaphase), failing to enter anaphase. Such a description had better be added in the result.

Answer: We thank the editor for their critical suggestion. Indeed, this suggestion is very helpful and relevant to describe the result of present study that the increased mitotic index and mitotic defect while at the same Cyclin B levels were elevated suggesting a significant delay of early mitotic phase. This information will be added as per this suggestion. Therefore, please kindly refer to our result section

Updated Result Section, please see page 14, line 451-459.

Inhibition of AURKB activated the phosphorylation of several cell cycle regulators; including Chk1, Cdc2, and Cyclin B1 (Figure 6D). AURKB inhibition resulted in higher mitotic indexes, as revealed by Giemsa staining, and mitotic defects, as revealed by fluorescence staining (Figure 6B, 6C). However, Cyclin B1 protein levels continued to increase (Figure 6D), suggesting that the early mitotic phase was delayed. Moreover, enhanced mitotic defects during mitosis progression in HNSCC cells were indicative of mitotic catastrophe triggered by mitosis delay. Hence, the findings indicate the high vulnerability of platinum-tolerant persister HNSCC cells to mitotic catastrophe upon inhibition of AURKB expression.

  1. line-459. “However, potential … to be determined.” This sentence should be described in the Introduction, as this makes a major motivation of the present study.

Answer: We thank the editor for their positive advice. According to this comment, we then re-described this sentence in the introduction section as one of the background of conducting this study. Therefore, please kindly refer to our Introduction section

Updated Introduction Section, please see page 3, line 119-126.

Herein, we examined the putative role of KDM5D in orchestrating AURKB expression might contribute the acquisition of DTPCs in HNSCC following platinum treatment. Platinum-tolerant persister cells of HNSCC then could be exploited by targeting KDM5D-associated control of cell cycle, DNA damage repair mechanism, and AURKB-mediated mitotic control by treating with AURKB inhibition, which provoked mitotic delay and ultimately resulted in mitotic catastrophe. Moreover, the clinical relevance of KDM5D as potential marker of DTPCs and platinum tolerance in HNSCC patients would be determined.

  1. line-464. “Diapause is a …development.” This sentence should be described in the Introduction, where the word “diapause” first appears.

Answer: We thank the editor for their critical suggestion. Indeed, this term is one of important mechanism in defining the biological properties of treatment-tolerant persister cancer cells and closely relevant with the result of present study. To enlighten any obscurity regarding this specific term and according to this comment, the description of this term will be provided and introduced. Therefore, please kindly refer to our Introduction section

Updated Introduction Section, please see page 2, line 66-72.

Cancer cells exhibit resistance to therapeutic drugs through various mechanisms. In drug-susceptible tumors, a small proportion of cells can transform into drug-tolerant persister cells (DTPCs) and contribute to the development of a drug-resistant population [8, 9]. DTPCs have distinct stemness characteristics, slow-cycling profile, quiescent behavior, and diapause-state-like properties [10, 11]. The diapause state represents a dormant, non-proliferating subset of cancer cells, mimicking embryonic development to prevent an exogenous insult, in this case, chemotherapy-induced cell death [12].

  1. line-473 to 475. “Histone demethylases … and pluripotency.” This basic information should be described in the Introduction.

Answer: We thank the editor for their critical suggestion. Indeed, this term is one of important mechanism in treatment-tolerant persister cancer cells and closely relevant with the result of present study. To enlighten any obscurity regarding this specific term and according to this comment, the description of this term will be provided and introduced. Therefore, please kindly refer to our Introducttion section

Updated Introduction Section, please see page 2, line 84-90.

Histone methylation can alter various biological features of tumors and serve as a potential target for eliminating treatment resistance [17, 18]. Thus, several histone demethylases have gained attention because they mainly play a key role in determining sensitivity following some types of treatment and because such misregulation can be targeted to overcome the development of treatment tolerance in cancer [16, 19, 20]. Histone demethylases regulate biological processes, such as cell cycle control, DNA damage responses, heterochromatin formation, and pluripotency [21].

  1. line-485 to 486. “suggesting that KDM5D … persister cells.” In the Discussion, discussions about possible molecular mechanisms how KDM5D provides cells of the treatment-tolerance have to be added, based on its molecular characteristics as a histone demethylase and the published knowledge on KDM proteins.

Answer: We thank the editor for this critical yet positive comment. Indeed, this suggestion is important to provide scientific explanation regarding the putative mechanism of KDM5D in regulating and promoting generation of treatment-tolerant persister cancer cells. To decipher the possible mechanism of our target, we then provided scientific explanation and background regarding the underlying mechanism of KDM5D to contribute for acquisition of DTPCs. Basic and relevant knowledge regarding KDM family will be also provided in introduction section. Therefore, please kindly refer to our Introduction and Discussion section

Updated Introduction Section, please see page 2-3, line 84-126.

Histone methylation can alter various biological features of tumors and serve as a potential target for eliminating treatment resistance [17, 18]. Thus, several histone demethylases have gained attention because they mainly play a key role in determin-ing sensitivity following some types of treatment and because such misregulation can be targeted to overcome the development of treatment tolerance in cancer [16, 19, 20]. Histone demethylases regulate biological processes, such as cell cycle control, DNA damage responses, heterochromatin formation, and pluripotency [21]. Among histone demethylases, KDM5 family have attracted significant contribution in cancer biology pertaining to acquisition of DTPCs. KDM5 family members are histone lysine deme-thylases that remove tri- and di-methyl marks from lysine residue (K4) of histone H3 protein (H3K4). Transcriptional regulation of the KDM5 family is either activated or repressed according to the site of methylation [22]. It has been shown, for instance, that subset of melanoma cells with aberrant KDM5B are likely surviving platinum treatment by transforming into a slow-cycling persister state [23]. By controlling chromatin marks of H3K4me3, KDM5 family could regulate the expression of mitotic-regulating genes, Aurora Kinase B (AURKB) [23, 24]. As a catalytic subunit of the chromosome passenger complex (CPC) during mitosis, AURKB facilitates chromosome alignment in metaphase and during cytokinesis [25]. Cancer cells could gain an advantage by modifying AURKB in a manner similar to how it functions during mitosis. Therefore, drugs targeting AURKB have become increasingly significant in recent years due to its potential to disrupt mitotic control of cancer cells while triggering a lethal cell death due to mitotic failure namely mitotic catastrophe [26]. Moreover, AURKB expression has been identified as a prognostic marker in several cancers, including oral cancer [27]. In this way, studying the role of KDM5 family members in maintaining DTPCs through exploiting certain cell cycle and mitosis control could yield an alternative method for identifying yet eliminating cancer cells associated with treatment refractoriness.

Among all KDM5 family members, KDM5D has received relatively less attention. KDM5D is frequently mutated in clear-cell renal cell carcinoma and is a major con-tributor to carcinogenesis [28]. KDM5D expression in gastric cancer cells substantially reduced these cells’ viability, implying that it may inhibit direct growth [29]. KDM5D deficiency results in increase of H3K4me3 methylation, leading to DNA replication stress and genomic instability [30]. This alteration increases the level of the G2/M checkpoint regulator and modulates the activation of the ATR-dependent mechanism through DNA replication stress. Thus, KDM5D is closely related to the epigenetic reg-ulation of cell cycle control in cancer. However, the significance of KDM5D to the de-velopment of DTPCs in HNSCC remains poorly understood. Herein, we examined the putative role of KDM5D in orchestrating AURKB expression might contribute the ac-quisition of DTPCs in HNSCC following platinum treatment. Platinum-tolerant per-sister cells of HNSCC then could be exploited by targeting KDM5D-associated control of cell cycle, DNA damage repair mechanism, and AURKB-mediated mitotic control by treating with AURKB inhibition, which provoked mitotic delay and ultimately resulted in mitotic catastrophe. Moreover, the clinical relevance of KDM5D as potential marker of DTPCs and platinum tolerance in HNSCC patients would be determined.

Updated Discussion Section, please see page 18, line 575-585.

Our study suggests that KDM5D regulates persister head and neck cancer cells by modulating AURKB expression. KDM5D is a lysine-specific demethylase that alters gene expression associated with cell cycle control and mitotic regulation by demethylating H3K4me3 and H3K4me2, which are also referred to as transcriptionally active chromatin [23]. In this case, we speculated that demethylation of H3K4me3 by KDM5D might lead to aberrant mRNA expression levels of mitotic-associated AURKB genes during platinum tolerance acquisition in HNSCC. As demonstrated in other studies, the demethylase activity of the KDM5 family is capable of activating cell cycle gene expression by determining H3K4me3 methylation levels at certain promoters [34]. Additionally, H3K4me3 demethylation has been shown to affect AURKB and E2F2 transcription levels in breast cancer tumors and is correlated with poor clinical outcomes [35].

  1. line-499. “as indicated by decreased gH2AX expression.” This interesting point should be more discussed, for example, by explicitly asking whether KDM5D makes the HNSCC persister cells insensitive to DNA damage (DNA damage exists but cells poorly sense it), invulnerable to DNA damaging (DNA damage does not occur in persister cells), or gain the increased DNA repair capability (DNA damage occurs but is more quickly repaired in persister cells than in normal cells).

Answer: We thank the editor for this suggestive and insightful comment. According to several possible assumption raised by this comment, we speculated that KDM5D might transform the HNSCC cells less vulnerable to DNA Damage because of several reasons that we will explain more in the main text. Therefore, please kindly refer to our Discussion section

Updated Discussion Section, please see page 17, line 559-569.

Moreover, our finding demonstrates that KDM5D protected HNSCC persister cells from DNA double-strand damage caused by cisplatin treatment, as indicated by pro-nounced γH2AX expression after KDM5D silencing. This finding indicated that KDM5D might gain the DNA repair capability of platinum-tolerant persister cells. This notion was in line with previous study that demonstrated KDM5D was involved in regulation of ATR-dependent DNA damage repair of prostate cancer as shown by in-creasing of CHK1 and CDC25c protein in response to KDM5D knockdown [22]. Similarly, our finding also noted increase phosphorylation of CHK1 and CDC25c after inhibition of AURKB by barasertib treatment, which mimicked the phenotypical response of KDM5D knockdown in platinum-tolerant HNSCC cells.

  1. line-518 to 520. Aurora kinases and their significance in cell division had better be described in the Introduction, as the present manuscript mentions Aurora kinases in the Abstract but not in the Introduction.

Answer: We thank the editor for this constructive comment. Indeed, Aurora Kinase is one of the important target and constructing the main axis with KDM5D in HNSCC cells, which should be explained well. To clarify the relevant role of Aurora Kinase in present study, some background related to functional aspect of AURKB will be provided and described, according to this suggestion. Therefore, please kindly refer to our Introduction section

Updated Introduction Section, please see page 2-3, line 97-109.

By controlling chromatin marks of H3K4me3, KDM5 family could regulate the expression of mitotic-regulating genes, Aurora Kinase B (AURKB) [23, 24]. As a catalytic subunit of the chromosome passenger complex (CPC) during mitosis, AURKB facilitates chromosome alignment in metaphase and during cytokinesis [25]. Cancer cells could gain an advantage by modifying AURKB in a manner similar to how it functions during mitosis. Therefore, drugs targeting AURKB have become increasingly significant in recent years due to its potential to disrupt mitotic control of cancer cells while triggering a lethal cell death due to mitotic failure namely mitotic catastrophe [26]. Moreover, AURKB expression has been identified as a prognostic marker in several cancers, including oral cancer [27]. In this way, studying the role of KDM5 family members in maintaining DTPCs through exploiting certain cell cycle and mitosis control could yield an alternative method for identifying yet eliminating cancer cells associated with treatment refractoriness.

  1. line 522. “This study” had better be “The present study,” to avoid confusion.

Answer: We thank the editor for their positive advice. According to this comment, we then replaced this sentence into other sentence as being suggested to reduce any confusion. Therefore, please kindly refer to our Discussion section

Updated Discussion section, please see page 18, line 597-599.

Treatment-tolerant persister cells are relatively susceptible to ferroptosis induction, and this susceptibility can be used to suppress the development of DTPCs [35-37]. The present study indicates alternative mechanisms for exploiting persister HNSCC cells, such as targeting mitotic catastrophe.

Reviewer 2 Report

This study, “KDMD5 histone demethylase identifies platinum-tolerant head and neck cancer cells vulnerable to mitotic catastrophe”, by Chen et al., reports overexpression of histone demethylase, KDMD5 in cisplatin-tolerant persistent Head and Neck Squamous Cell Carcinoma (HNSCC) cells and present KDMD5 as a predictive marker for recurrence and relapse of HNSCC after platinum therapy. Furthermore, Chen et al., report overexpression of Aurora kinase B (AURKB) by KDMD5 and involvement of KDMD5/AURKB axis in development of HNSCC-persister cells. Finally, the authors have demonstrated mitotic catastrophe in cisplatin-persister HNSCC cells after AURKB disruption/inhibition and suggest it as a therapeutic approach to increase vulnerability of persister cells and potentially overcome platinum tolerance in HNSCC.

Major revisions:

(i) Fig 1A: Is it possible to show exact location of KDMD5 in each of the three volcano plots by a different color.  Fig 1C: AURKB is absent in this heat map. In addition to KDMD5, AURKB is another important gene discussed in this study, hence is there any specific reason, it is missing in this DEGs list. Fig 1E: MYC-E2F1-DNA signaling pathway is shown as upregulated in the figure but in the corresponding result section (2.1) it is mentioned as downregulated. Fig 1F: Names of few diapause related genes used in this figure can be mentioned in the corresponding result section.

(ii) Fig 3/Result 2.3: ALDH1A3, AURKB and KDMD5 are expressed in cluster 1,3 and 5 (high persister cells) but their expression is weak. Hence in the context of a weak expression of AURKB and KDMD5, what advantage these two genes have over other (high expression levels) markers in these clusters for determining tumor persistence. These genes have a role in cell cycle checkpoint and DNA damage. How a weak expression of these genes justify their role as predictive/diagnostic marker in comparison to other highly expressed genes in these persister cells, please discuss.

(iii) Fig 3E: Does expression of ALDH1A3, KDMD5, AURKB was also observed along with up regulation of a specific diapause gene in a single cell. It will be good to show any overlap/colocalization between ALDH1A3, KDMD5, AURKB with a specific diapause-related marker protein.  Additionally, a quantitative data graph can better represent the correlation between KDMD5/AURKB/ALDH1A3 expression and upregulation of diapause-related gene (s).

(iv) Result 2.3: Line 244-246: Furthermore, relatively abundant ALDH1A3 and KDM5D expression in cell clusters 1, 3, and 5 resulted in the upregulation of the genetic signatures of the diapause state. Which evidence suggests that ALDH1A3 and KDMD5 cause upregulation of diapause related genes?

(v) Fig 4: None of the figures/results show that KDMD5 generates platinum-tolerant persister cells through AURKB upregulation as mentioned in lines 295-296. Fig 4B shows decrease in relative mRNA expression of AURKB upon KDMD5 knockdown in the two platinum-persister cell lines, but this data is not sufficient to make this statement. Upregulation of AURKB as a direct response to overexpressed KDMD5 needs to be shown. Fig 4C: Missing methodology for tumor sphere formation. Either provide methodology or the reference.

(vi) Fig 5A: There is no change in colony number upon KDMD5 knockdown in platinum-persister cells in absence of cisplatin (0 mg/ml) as compared to scrambled control. Usually, knockdown of KDMD5 is also an external treatment/intervention, hence any reason why knockdown of a gene responsible for platinum-tolerance does not reduce colony number?

(vii) Fig 5E: In PT-SAS cell line, substantial γH2AX expression is visible in Sh-NC positive image. Hence, qualitatively the effects of KDMD5 knockdown are not well defined. Perhaps, quantitative graphical analysis can better show the enhanced expression of DNA damage marker, γH2AX after KDMD5 knockdown.

(viii) Fig 6B: The general trend in this figure is of increase in mitotic index with increase in Barasertib concentration for each time point for each cisplatin tolerant persister HNSCC cell line. Since Barasertib, inhibits cell proliferation, cell division and induce apoptosis in AURKB overexpressing tumor cells and as authors suggested AURKB inhibition as a potential treatment to overcome platinum tolerance in HNSCC cells, it will be helpful to discuss further why Barasertib induced AURKB inhibition is causing increase in mitosis even after 48 hrs post Barasertib treatment in cisplatin-tolerant persister HNSCC cells and if this has any effects on cell viability, proliferation and colony number.

(ix) Statistical methodology is missing and needs to be included in methodology.

(x) A positive control is missing to compare results obtained for KDMD5 and AURKB overexpression/knockdown with an established positive marker for cancer stemness, cisplatin resistance and diapause.

Minor revisions:

(i) Some grammatical and spelling mistakes.

(ii) Scale bar needs to be mentioned in figure legends wherever applicable.

(iii) Fig 1A: What does NS, FC, P, FC_P represent, needs to be mentioned in the fig legend.

(iv) Fig 1E: What does NES represent, needs to be mentioned in fig legend.

Author Response

Reviewer #2: This study, “KDMD5 histone demethylase identifies platinum-tolerant head and neck cancer cells vulnerable to mitotic catastrophe”, by Chen et al., reports overexpression of histone demethylase, KDMD5 in cisplatin-tolerant persistent Head and Neck Squamous Cell Carcinoma (HNSCC) cells and present KDMD5 as a predictive marker for recurrence and relapse of HNSCC after platinum therapy. Furthermore, Chen et al., report overexpression of Aurora kinase B (AURKB) by KDMD5 and involvement of KDMD5/AURKB axis in development of HNSCC-persister cells. Finally, the authors have demonstrated mitotic catastrophe in cisplatin-persister HNSCC cells after AURKB disruption/inhibition and suggest it as a therapeutic approach to increase vulnerability of persister cells and potentially overcome platinum tolerance in HNSCC.

Major revisions:

  1. Fig 1A: Is it possible to show exact location of KDMD5 in each of the three volcano plots by a different color. Fig 1C: AURKB is absent in this heat map. In addition to KDMD5, AURKB is another important gene discussed in this study, hence is there any specific reason, it is missing in this DEGs list. Fig 1E: MYC-E2F1-DN signaling pathway is shown as upregulated in the figure but in the corresponding result section (2.1) it is mentioned as downregulated. Fig 1F: Names of few diapause related genes used in this figure can be mentioned in the corresponding result section.

Answer: We thank the editor for their positive advice.

  • For figure 1A, the dot position of KDM5D was not shown in our previous draft. According to this advice, we then showed the exact position of KDM5D in each volcano plot by red dot along with arrow. Therefore, please kindly refer to our updated Figure 1A.
  • For figure 1C, AURKB was absent in this heat map because in our previous approach we could not find AURKB as the DEGs between CSC and non-CSC subset. However, we introduced AURKB in the re-analysis of single-cell RNA-sequencing dataset, in which the scatter plot showed that KDM5D was significantly correlated with AURKB expression in figure 3E. We then also confirmed that silencing of KDM5D reduced AURKB mRNA level in figure 4B, suggesting that KDM5D regulate AURKB expression and indicating the presence of KDM5D/AURKB axis activation in PT-SAS and PT-FADU cells. The selection of AURKB was previously speculated from the literature observation that demonstrated KDM5 family could regulate cell cycle and mitotic activity of drug-tolerant persister cancer cells (DTPCs) by altering AURKB expression. However, those studies were previously reported only in KDM5A and KDM5B, while the function of KDM5D in regulating AURKB is still unclear. Moreover, targeting AURKB was also recently gaining some interest to eliminate drug-tolerant persister cells. In line with this information, our in silico dataset re-analysis and IHC staining of HNSCC patients also confirmed high expression of KDM5D was associated with platinum sensitivity (Figure 1A, 1G, 2B, 2D, and 2E), therefore we were then curiously investigate whether KDM5D might also regulate AURKB then promote platinum tolerance in HNSCC tumor which was indeed present. To reduce any gaps of how we determine AURKB, we then also provided key information regarding KDM5 regulation to AURKB level and recent significance of AURKB in the introduction section. Therefore, please kindly refer to our Introduction section.
  • For figure 1E, MYC_E2F1_DN signaling pathway is shown as upregulated in the figure but in the corresponding result section we described it as downregulated. This is because MYC_E2F1_DN signaling pathway consisted genes that are downregulated in cancer cells with activation of MYC_E2F1 pathway. Therefore, in our case, due the enrichment score was positive for MYC_E2F1_DN signaling pathway (NES = 1.3), this result suggested that MYC_E2F1 signalling pathway was deactivated or downregulated in CSC than non-CSC cells.
  • For figure 1F, according to this suggestion, we will provide several genes that construct this gene set signature in the corresponding result section. Therefore, please kindly refer to our Result section.

Updated figure 1A, please see page 5.

Updated legend of figure 1A, please see page 5.

Figure 1. Differentially expression of KDM5D linked HNSCC Tumors, CSC, and Cisplatin Resistance. (A) Individual volcano plot depicted overexpression of KDM5D as DEGs in respective phenotypes and datasets: Tumor vs Normal (GSE9844), CSC vs Non-CSC (GSE72384), and Cisplatin Resistant vs Sensitive (GSE102787). The position of KDM5D was marked in red dot with respective arrow.

Updated Introduction Section, please see page 2-3, line 84-109.

Histone methylation can alter various biological features of tumors and serve as a potential target for eliminating treatment resistance [17, 18]. Thus, several histone demethylases have gained attention because they mainly play a key role in determining sensitivity following some types of treatment and because such misregulation can be targeted to overcome the development of treatment tolerance in cancer [16, 19, 20]. Histone demethylases regulate biological processes, such as cell cycle control, DNA damage responses, heterochromatin formation, and pluripotency [21]. Among histone demethylases, KDM5 family have attracted significant contribution in cancer biology pertaining to acquisition of DTPCs. KDM5 family members are histone lysine demethylases that remove tri- and di-methyl marks from lysine residue (K4) of histone H3 protein (H3K4). Transcriptional regulation of the KDM5 family is either activated or repressed according to the site of methylation [22]. It has been shown, for instance, that subset of melanoma cells with aberrant KDM5B are likely surviving platinum treatment by transforming into a slow-cycling persister state [23]. By controlling chromatin marks of H3K4me3, KDM5 family could regulate the expression of mitotic-regulating genes, Aurora Kinase B (AURKB) [23, 24]. As a catalytic subunit of the chromosome passenger complex (CPC) during mitosis, AURKB facilitates chromosome alignment in metaphase and during cytokinesis [25]. Cancer cells could gain an advantage by modifying AURKB in a manner similar to how it functions during mitosis. Therefore, drugs targeting AURKB have become increasingly significant in recent years due to its potential to disrupt mitotic control of cancer cells while triggering a lethal cell death due to mitotic failure namely mitotic catastrophe [26]. Moreover, AURKB expression has been identified as a prognostic marker in several cancers, including oral cancer [27]. In this way, studying the role of KDM5 family members in maintaining DTPCs through exploiting certain cell cycle and mitosis control could yield an alternative method for identifying yet eliminating cancer cells associated with treatment refractoriness.

Among all KDM5 family members, KDM5D has received relatively less attention. KDM5D is frequently mutated in clear-cell renal cell carcinoma and is a major contributor to carcinogenesis [28]. KDM5D expression in gastric cancer cells substantially reduced these cells’ viability, implying that it may inhibit direct growth [29]. KDM5D deficiency results in increase of H3K4me3 methylation, leading to DNA replication stress and genomic instability [30]. This alteration increases the level of the G2/M checkpoint regulator and modulates the activation of the ATR-dependent mechanism through DNA replication stress. Thus, KDM5D is closely related to the epigenetic regulation of cell cycle control in cancer. However, the significance of KDM5D to the development of DTPCs in HNSCC remains poorly understood. Herein, we examined the putative role of KDM5D in orchestrating AURKB expression might contribute the acquisition of DTPCs in HNSCC following platinum treatment. Platinum-tolerant per-sister cells of HNSCC then could be exploited by targeting KDM5D-associated control of cell cycle, DNA damage repair mechanism, and AURKB-mediated mitotic control by treating with AURKB inhibition, which provoked mitotic delay and ultimately resulted in mitotic catastrophe. Moreover, the clinical relevance of KDM5D as potential marker of DTPCs and platinum tolerance in HNSCC patients would be determined.

Updated Result section, please see page 4, line 167-173.

Therefore, by employing the profiling results of the TCGA-HNSC dataset, we examined the correlation between diapause enrichment and KDM5D expression. Previous study has already established 14 genes that were upregulated (e.g.: PDCD4, ACSS1, HEXB, CTSL, CCNG2, SPRY1, APOE, ALDH6A1) and 110 genes that were downregulated (e.g.: LDHA, PPA1, S100A6, ID3, MGMT, PRDX2, HSPE1, EIF2B2, CENPM, DRG2, PDCD5, EIF3B, CCDC28B, BRMS1) during embryonal diapause state [11]. The scatter plot revealed that KDM5D expression was positively correlated with genes upregulated in the diapause-like state (r = 0.22, P = 0.0032) and negatively correlated with genes downregulated in the diapause-like state (r = −0.26, P = 0.0077; Figure 1F).

  1. Fig 3/Result 2.3: ALDH1A3, AURKB and KDMD5 are expressed in cluster 1,3 and 5 (high persister cells) but their expression is weak. Hence in the context of a weak expression of AURKB and KDMD5, what advantage these two genes have over other (high expression levels) markers in these clusters for determining tumor persistence. These genes have a role in cell cycle checkpoint and DNA damage. How a weak expression of these genes justify their role as predictive/diagnostic marker in comparison to other highly expressed genes in these persister cells, please discuss.

Answer: We thank the editor for this critical comment. Regarding the relatively low expression of KDM5D, AURKB, and ALDH1A3 in figure 3B, this principally occurred due to the fact that this dataset (GSE103322) was originated from the fresh biopsy samples of HNSCC patients in which subsequent isolation or cell-sorting of specific DTP or CSC cells were not performed. Thus, the percentage of DTP or CSC cells were relatively minor (as also implied by relatively low CSC marker such as ALDH1A3). On the other hand, representative marker of HNSCC cells such as KRT6A, KRT14, and CDH1 were relatively abundant and consistently reflected that the biopsy samples were HNSCC tumors. However, we could demonstrate significance expression of KDM5D in our IHC staining particularly among HNSCC patients who were not responding to platinum treatment or suffering early recurrence. This indicated that KDM5D, AURKB, and ALDH1A3 were context-dependent and subset-specific (higher in subset of cells that were likely becoming platinum-tolerant). In our humble opinion, the key result of this figure is the significant correlation between KDM5D, AURKB, and ALDH1A3, which is related to diapause state activation, not the minor percentage expression of those genes in the overall HNSCC sample. Then, the predictive/diagnostic value of KDM5D among HNSCC patients were concluded by our result in Table 1 in which higher KDM5D expression was associated with poor response of platinum treatment (p = 0.004) and early recurrence (p = 0.012). To explain the main reason of the weak expression of those genes, several description were then provided in the corresponding result section. Therefore, please kindly refer to our Result section

Updated Result section, please see page 7-8, line 263-270.

An established cancer stemness marker was used in the present study, namely ALDH1A3 since this Aldehyde Dehydrognease isoenzyme has also recently identified to be enriched in drug-tolerant persister cancer cells, cisplatin resistance, and radiore-sistance of head and neck cancer [33-35]. Here, we noted a weak expression of specific cancer stem cell markers and DTPC markers, such as ALDH1A3, in some clusters, such as clusters 1, 3, and 5 (Figure 3C). Relatively overall weak expression of cancer stem cell marker reflected the sampling approach of the dataset, which was derived from the fresh biopsy pretreatment HNSCC tumors without further isolation or enrichment to CSC subset.

  1. Fig 3E: Does expression of ALDH1A3, KDMD5, AURKB was also observed along with up regulation of a specific diapause gene in a single cell. It will be good to show any overlap/colocalization between ALDH1A3, KDMD5, AURKB with a specific diapause-related marker protein. Additionally, a quantitative data graph can better represent the correlation between KDMD5/AURKB/ALDH1A3 expression and upregulation of diapause-related gene (s).

Answer: We appreciate the editor for receiving a critical comment. Indeed, providing additional figure to describe overlap expression between several diapause-related genes and KDM5D would more convince presumptive function of KDM5D in controlling diapause state. Several diapause markers were selected from previous studies, such as CCND1 (Cyclin D1), FAS, and ALDH6A1. Scatter plots, which showed correlation between KDM5D and the expression of those diapause markers, were also provided. Therefore, please kindly refer to our updated figure 3B,C,E and respective result section.

Updated Figure 3, please see page 9.

Updated Legend of Figure 3, please see page 9, line 300-310.

Figure 3. KDM5D and AURKB co-expression delineates cluster of HNSCC persister cells. (A) Representative tSNE plots of single-cell profiling in GSE103322 dataset showed eight distinct cluster of cells. (B) Array of tSNE plots portrayed expression level of interest genes such as HNSCC tumor markers (KRT6A, KRT14, CDH1), CSC and persister marker (ALDH1A3), putative main targets of this study (KDM5D and AURKB), and diapause-related genes (CCND1, FAS, ALDH6A1). (C) Dot plot described level of expression of each gene (KDM5D, ALDH1A3, AURKB, CCND1, FAS, ALDH6A1) in eight distinct clusters. (D) Scatter plot depicted co-association of each interest genes; pearson’s coefficient and p-value was provided in the top margin. The KDM5D expression was significantly correlated to AURKB (r = 0.75, P = 0.012) and ALDH1A3 (r = 0.38, P = 0.035). (E) Scatter plot portrayed correlation between several diapause-related genes and KDM5D, such as CCND1 (r = 0.21, P = 0.037), FAS (r = 0.24, P = 0.021), and ALDH6A1 (r = 0.39, P = 0.018).

Updated Result section, please see page 8, line 281-291.

To confirm our speculation regarding association between KDM5D expression with diapause state, several markers that were upregulated during diapause were se-lected from previous report, such as Cyclin D1 (CCND1), Fatty Acid Synthase (FAS), and ALDH6A1. The level of expression of those markers were also shown in each clus-ter (figure 3B, C). While overlap expression between KDM5D and these markers was not entirely evident, correlation analysis resulted in significant associations between KDM5D and CCND1, FAS, and ALDH6A1. (Figure 3E). Moreover, the relatively abundant expression of KDM5D and ALDH1A3 in cell clusters 1, 3, and 5 were over-lapped with a portion of clusters upregulating diapause gene signatures (Dia-pause_UP). Moreover, these clusters of cells (no. 1, 3, and 5) co-existed with clusters that minimally expressed genes that consistently deactivated during diapause (Dia-pause_DOWN) (figure 3F).

  1. Result 2.3: Line 244-246: Furthermore, relatively abundant ALDH1A3 and KDM5D expression in cell clusters 1, 3, and 5 resulted in the upregulation of the genetic signatures of the diapause state. Which evidence suggests that ALDH1A3 and KDMD5 cause upregulation of diapause related genes?

Answer: We appreciate the editor for receiving a critical comment. After we digest this sentence again, we realized this sentence was somehow over claim as our data did not provide evidence regarding the role of KDM5D in regulating diapause related genes expression. The sentences were then reconstructed to reduce any bias and misunderstanding. Therefore, please kindly refer to our updated Result section.

Updated Result section, please see page 8, line 287-293.

Moreover, the relatively abundant expression of KDM5D and ALDH1A3 in cell clusters 1, 3, and 5 were overlapped with a portion of clusters upregulating diapause gene signatures (Diapause_UP). Moreover, these clusters of cells (no. 1, 3, and 5) co-existed with clusters that minimally expressed genes that consistently deactivated during diapause (Diapause_DOWN) (figure 3F). These findings indicated that the clusters contained a substantial number of cells in the diapause state, a key feature of DTPCs in HNSCC.

  1. Fig 4: None of the figures/results show that KDMD5 generates platinum-tolerant persister cells through AURKB upregulation as mentioned in lines 295-296. Fig 4B shows decrease in relative mRNA expression of AURKB upon KDMD5 knockdown in the two platinum-persister cell lines, but this data is not sufficient to make this statement. Upregulation of AURKB as a direct response to overexpressed KDMD5 needs to be shown. Fig 4C: Missing methodology for tumor sphere formation. Either provide methodology or the reference.

Answer: We thank the editor for their positive advice.

  • For figure 4B and respective result section, after we digest this sentence again, we realized this sentence was bias as our data did not provide evidence regarding upregulation of AURKB by KDM5D. The sentences were then replacing this sentence then reconstructed it to reduce any bias and misunderstanding. Therefore, please kindly refer to our updated Result section.
  • For figure 4C, we then added the information pertaining to the tumorsphere formation assay in the method section. Therefore, please kindly refer to our Methods section.

Updated Result Section, please see page 10, line 354-357.

Overall, the data highlight KDM5D/AURKB axis in which KDM5D modulates AURKB expression, generates platinum-tolerant persister cells, enhances cancer stemness potential, activates the diapause-like state, and alters platinum sensitivity in HNSCC.

Updated Methods Section, please see page 20, line 689-696.

4.9. Tumorsphere Formation

The platinum-tolerant HNSCC cells (PT-SAS and PT-FaDu) were seeded in se-rum-free low-adhesion culture plates containing of stem cell media which the content as follows: RPMI1640 with B27 supplement (Invitrogen), 20 ng/mL EGF, and 20 ng/mL basic-FGF (stem cell medium; PeproTech, Rocky Hill, NJ, USA). The cells were grown for about 14 days to allow formation of spheres. The spheres were then counted under a microscope and spheres formation efficiency was calculated as the ratio of the num-ber of spheres formed to the seeded adherent cell number.

  1. Fig 5A: There is no change in colony number upon KDMD5 knockdown in platinum-persister cells in absence of cisplatin (0 mg/ml) as compared to scrambled control. Usually, knockdown of KDMD5 is also an external treatment/intervention, hence any reason why knockdown of a gene responsible for platinum-tolerance does not reduce colony number?

Answer: We appreciate the editor for receiving this critical comment. In our previous result, relative colony number was used, in which each untreated cells (either shNC or shKDM5D) was normalized to 100%. This method however seems to make confusion regarding the effect of KDM5D knockdown in the untreated colony. Hence, we then re-counted our colony formation result and made the untreated shNC as the 100% relative control for all other groups, in which indeed shKDM5D group reduced colony formation than shNC. Therefore, please kindly refer to our updated Figure 5A.

Updated Figure 5A, please see page 13.

  1. Fig 5E: In PT-SAS cell line, substantial γH2AX expression is visible in Sh-NC positive image. Hence, qualitatively the effects of KDMD5 knockdown are not well defined. Perhaps, quantitative graphical analysis can better show the enhanced expression of DNA damage marker, γH2AX after KDMD5 knockdown.

Answer: We appreciate the editor for receiving this constrictive advice. Indeed, the addition of quantitive graphical analysis will help any audience to understand the marker alteration during specific treatment. Hence, we then generated the graphical analysis for γH2AX and EdU staining in our updated figure. Therefore, please kindly refer to our updated Figure 5E, and 5F.

Updated Figure 5E and 5F, please see page 13.

Updated Legends of Figure 5E,F, please see page 13-14, line 422-426.

(E) The level of DNA damage upon cisplatin treatment was significantly increased in response to KDM5D silencing, as indicated by the percentage of γH2AX positive cells with >10 foci in both PT-SAS and PT-FaDu cells. (F) Cellular proliferation was significantly reduced following KDM5D silencing and become more suppressed after cisplatin treatment, as reflected by decrease of EdU-positive cells fraction in both PT-SAS and PT-FaDu cells.

Updated Result section, please see page 12, line 399-412.

To determine whether KDM5D alters the rate of DNA damage, we suppressed KDM5D expression and examined the extent of DNA damage by quantifying γH2AX, a well-known immunofluorescence marker for localizing DNA damage [34]. Following platinum-DNA adduct formation and the recruitment of DNA repair foci in response to double-strand breaks, fluorescent subnuclear foci were detected by H2AX immunofluorescence staining. Moreover, we determined the proliferation rate following cisplatin-induced DNA damage through EdU fluorescence staining. Representative images of γH2AX and EdU staining in both PT-SAS and PT-FaDU cells were shown in Figure 5G. In the present study, KDM5D inhibition markedly increased the expression of γH2AX during platinum treatment while EdU expression was significantly reduced in both PT-SAS and PT-FaDu cells (Figure 5E, F). The findings suggest higher extent of DNA damage in response to KDM5D silencing, reflecting the protective role of KDM5D to prevent DNA damage while maintaining proliferation of persister HNSCC cells following cisplatin treatment.

  1. Fig 6B: The general trend in this figure is of increase in mitotic index with increase in Barasertib concentration for each time point for each cisplatin tolerant persister HNSCC cell line. Since Barasertib, inhibits cell proliferation, cell division and induce apoptosis in AURKB overexpressing tumor cells and as authors suggested AURKB inhibition as a potential treatment to overcome platinum tolerance in HNSCC cells, it will be helpful to discuss further why Barasertib induced AURKB inhibition is causing increase in mitosis even after 48 hrs post Barasertib treatment in cisplatin-tolerant persister HNSCC cells and if this has any effects on cell viability, proliferation and colony number.

Answer: We appreciate the editor for receiving this critical comment. Indeed, this question was also similarly raised with the other review who speculated delay of mitosis progression upon barasertib treatment. The delay was particularly occurred in the early phase of mitosis since we found the increase of mitotic index was followed by upregulation of several mitotic regulator markers such as Cdc2 and Cyclin B1 protein. However, this mitosis delay is relevant since delay of mitosis progression would consequently lead into lethal mitotic catastrophe as we also found increase of mitotic defect by observing through Giemsa staining study. Hence, we then elaborate this explanation into result section. Therefore, please kindly refer to our updated Result section.

Updated Result section, please see page 14, line 451-459.

Inhibition of AURKB activated the phosphorylation of several cell cycle regulators; in-cluding Chk1, Cdc2, and Cyclin B1 (Figure 6D). AURKB inhibition resulted in higher mitotic indexes, as revealed by Giemsa staining, and mitotic defects, as revealed by fluorescence staining (Figure 6B, 6C). However, Cyclin B1 protein levels continued to increase (Figure 6D), suggesting that the early mitotic phase was delayed. Moreover, enhanced mitotic defects during mitosis progression in HNSCC cells were indicative of mitotic catastrophe triggered by mitosis delay. Hence, the findings indicate the high vulnerability of platinum-tolerant persister HNSCC cells to mitotic catastrophe upon inhibition of AURKB expression.

  1. Statistical methodology is missing and needs to be included in methodology.

Answer: We appreciate the editor for receiving this positive suggestion. As per suggestion, we then added statistical analysis in the methods section. Therefore, please kindly refer to our updated Methods section.

Updated Methods section, please see page 21-22, line 765-776.

4.15. Statistical Analysis

A mean and a standard error of the mean were calculated for each numerical variable. Frequency and percentage were used to represent categorical variables. The mean values between the two groups were compared by an unpaired Student's t test. Analysis of variance (ANOVA) was used to identify discrepancies between groups. In cases where ANOVA results were significant, the least significant difference test was employed to test for differences between groups. In order to compare different groups under different timelines, we applied a two-way ANOVA with repeated measures. In order to determine the correlation strength between the different parameters, Pear-son's linear correlation was applied. Statistical significance was determined by a p-value lower than 0.05. All tests were conducted in triplicate and analyzed using R studio (version 1.4.1717, Boston, MA, USA) and GraphPad Prism (version 8.02, San Diego, CA, USA).

  1. A positive control is missing to compare results obtained for KDMD5 and AURKB overexpression/knockdown with an established positive marker for cancer stemness, cisplatin resistance and diapause.

Answer: We appreciate the editor for receiving this critical comment. We have added this analysis to the Supplementary S1 Data and described in Results.

Updated Result section, please see page 12, line 410.

“2.5. KDM5D protects DNA damage following platinum treatment in persister cells

Platinum agents, such as cisplatin, carboplatin, and oxaliplatin, work by forming covalent binds with DNA, leading to the formation of DNA crosslinks and thus the in-hibition of DNA replication, arrest of the cell cycle, and cessation of cancer cell prolif-eration. To determine whether KDM5D affects sensitivity to platinum treatment, un-der either the suppression or overexpression of KDM5D, we examined the viability, cell cycle progression, and DNA damage of platinum-tolerant HNSCC cells following cisplatin treatment. The suppression of KDM5D expression increased the vulnerability of both PT-SAS and PT-FaDU cells against cisplatin, as indicated by a significant re-duction in the number of colonies upon cisplatin treatment (Figure 5A). Moreover, both HNSCC cell lines with KDM5D overexpression exhibited increased tolerance against several incremental dosages of cisplatin (Figure 5B). The results revealed that the tolerance of HNSCC cells to cisplatin treatment is mediated by KDM5D, and the expression of this gene increased tolerability to platinum agents.Cisplatin promotes cell cycle arrest in cancer cells. However, the increased tolerance of platinum-tolerant HNSCC cells diminishes this effect. The distribution of the cell cycle did not signifi-cantly change after cisplatin treatment in both PT-SAS and PT-FaDU cells (Figure 5C).

 Consistent with the previous result presented in Figure 4D, the inhibition of KDM5D through shRNA-mediated knockdown induced cell cycle arrest in PT-SAS and PT-FaDU cells, as indicated by an increase in the G1 cell subpopulation and a decrease in the S cell subpopulation (Figure 5D). After cisplatin treatment, cell cycle arrest was markedly enhanced after the knockdown of KDM5D. Thus, KDM5D appears to be a key factor determining the vulnerability to platinum agents by governing the cell cycle progression of persister HNSCC cells. Platinum agents can eliminate cancer cells by forming DNA crosslinks, resulting in DNA damage. The tolerability of persister HNSCC cells against platinum agents can be mediated by lowering their susceptibility to DNA damage. To determine whether KDM5D alters the rate of DNA damage, we suppressed KDM5D expression and examined the extent of DNA damage by quanti-fying γH2AX, a well-known immunofluorescence marker for localizing DNA damage [38]. Following platinum-DNA adduct formation and the recruitment of DNA repair foci in response to double-strand breaks, fluorescent subnuclear foci were detected by H2AX immunofluorescence staining. Moreover, we determined the proliferation rate following cisplatin-induced DNA damage through EdU fluorescence staining. Repre-sentative images of γH2AX and EdU staining in both PT-SAS and PT-FaDU cells were shown in Figure 5G. In the present study, KDM5D inhibition markedly increased the expression of γH2AX during platinum treatment while EdU expression was signifi-cantly reduced in both PT-SAS and PT-FaDu cells (Figure 5E, F). We also compare re-sults obtained for KDMD5 and AURKB overexpression/knockdown with an established positive marker for cancer stemness (SOX2), cisplatin resistance (Cyclins D1) and dia-pause(NRF2) as shown in Supplementary Figure S1. The findings suggest higher ex-tent of DNA damage in response to KDM5D silencing, reflecting the protective role of KDM5D to prevent DNA damage while maintaining proliferation of persister HNSCC cells following cisplatin treatment.”

To represent the positive marker of cancer stemness, cisplatin resistance, drug tolerance and diapause, we previously have already used ALDH1A3 since this aldehyde dehydrogenase isoenzyme has been known to possess multiple vital role in regulating cancer stemness and drug-tolerant persister cells. To provide clarification pertaining to the selection of ALDH1A3 and its role in this study, several information then would be added in the result section including several newly cited references. Therefore, please kindly refer to our updated Result section.

Updated Result section, please see page 10, line 323-341.

The platinum-tolerant cell lines were then used as basic model for dissecting the basic molecular mechanism of drug tolerance acquisition. Here, ALDH1A3 was again being used to remark enrichment of drug-tolerant persister cancer cells, cisplatin resistance, and quiescent population resembling diapause state [33-35, 37]. The quantitative polymerase chain reaction revealed relatively high expression levels of KDM5D, ALDH1A3, and AURKB in wild-type platinum-tolerant SAS (PT-SAS) and FaDu (PT-FaDU) cells (Figure 4B). According to previous in silico finding through multiple sets of transcriptomic profiling and our speculation regarding the putative role of KDM5D/AURKB axis in contributing drug-tolerant persister cells, then KDM5D silencing was performed in platinum-tolerant HNSCC cells and determine the down-stream modulation by KDM5D. As expected, knockdown of KDM5D significantly reduced the expression levels of AURKB and ALDH1A3 in both SAS and FaDu persister cells, indicating that KDM5D regulates AURKB and ALDH1A3 expression (Figure 4B). As such, the presence of KDM5D/AURKB axis was then confirmed and activated in platinum-tolerant HNSCC cells. Downregulation of ALDH1A3 in response to KDM5D silencing also suggested the role of KDM5D/AURKB axis in modulating ALDH1A3-mediated cancer stemness, plati-num tolerance, and transition towards quiescent state of HNSCC cells. Further characterization and functional perturbation assay were then examined to demonstrate functional role of KDM5D related to drug tolerance.

Additional Reference:

Cortes-Dericks, L., Froment, L., Boesch, R., Schmid, R.A., Karoubi, G., 2014. Cisplatin-resistant cells in malignant pleural mesothelioma cell lines show ALDHhighCD44+ phenotype and sphere-forming capacity. BMC Cancer 14, 304.

Durinikova, E., Kozovska, Z., Poturnajova, M., Plava, J., Cierna, Z., Babelova, A., Bohovic, R., Schmidtova, S., Tomas, M., Kucerova, L., Matuskova, M., 2018. ALDH1A3 upregulation and spontaneous metastasis formation is associated with acquired chemoresistance in colorectal cancer cells. BMC Cancer 18, 848.

Kawakami, R., Mashima, T., Kawata, N., Kumagai, K., Migita, T., Sano, T., Mizunuma, N., Yamaguchi, K., Seimiya, H., 2020. ALDH1A3-mTOR axis as a therapeutic target for anticancer drug-tolerant persister cells in gastric cancer. Cancer Science 111, 962-973.

Kurth, I., Hein, L., Mäbert, K., Peitzsch, C., Koi, L., Cojoc, M., Kunz-Schughart, L., Baumann, M., Dubrovska, A., 2015. Cancer stem cell related markers of radioresistance in head and neck squamous cell carcinoma. Oncotarget; Vol 6, No 33.

Minor revisions:

  1. Some grammatical and spelling mistakes.

Answer: We appreciate the editor for receiving this positive suggestion. Here, we have already tried our best to fully read again the whole draft and revise any grammatical with spelling mistakes.

  1. Scale bar needs to be mentioned in figure legends wherever applicable.

Answer: We appreciate the editor for receiving this constructive comment. In response to this advice, each scale bar of the figure has been added and updated regarding the relative size in the corresponding figure legends. Therefore, please kindly refer to our updated Legend of Figure 2, 4, 5, and 7.

Updated Figure Legend section, please see page 7,11,13,16.

Figure 2. Overexpression of KDM5D was associated to poor platinum responses in HNSCC pa-tients. (A) Representative … Scale bar: 200 μm,

Figure 4. KDM5D promotes generation of platinum-tolerant persister cells. (A) Brief schematic figure shows the steps to generate cisplatin-tolerant persister cells. … Scale bar: 100 μm.

Figure 5. KDM5D abrogates DNA damage and cell cycle arrest upon platinum treatment. (A) The suppression … Scale bar: 100 μm.

Figure 7. Cisplatin and barasertib co-treatment extends tumor suppression poten-tial in vivo. (A) Treatment … Scale bar: 200 μm.

  1. Fig 1A: What does NS, FC, P, FC_P represent, needs to be mentioned in the fig legend.

Answer: We appreciate the editor for receiving this constructive comment. As per this advice, the meaning of each legend text would then be described the corresponding figure legend. Therefore, please kindly refer to our updated Legend of Figure 1.

Updated Figure Legend section, please see page 5, line 194-196.

Figure 1. Differentially expression of KDM5D linked HNSCC Tumors, CSC, and Cisplatin Resistance. (A) Individual volcano plot depicted overexpression … NS: Not Significant, FC: Significant in Log2 Fold-Change, P: Significant in P-value, FC_P: Significant in Log2 Fold-Change and p-value, NES: Normalized Enrichment Score.

  1. Fig 1E: What does NES represent, needs to be mentioned in fig legend.

Answer: We appreciate the editor for receiving this constructive comment. As per this advice, the meaning of NES (Normalized Enrichment Score) would be described in the corresponding figure legend. Therefore, please kindly refer to our updated Legend of Figure 1.

Updated Figure Legend section, please see page 5, line 194-196.

Figure 1. Differentially expression of KDM5D linked HNSCC Tumors, CSC, and Cisplatin Resistance. (A) Individual volcano plot depicted overexpression … NS: Not Significant, FC: Significant in Log2 Fold-Change, P: Significant in P-value, FC_P: Significant in Log2 Fold-Change and p-value, NES: Normalized Enrichment Score.
